# Individualized prevention of proton pump inhibitor related adverse events by risk stratification

Bin Xia[1,2,17], Qiangsheng He[1,2,17], Fang Gao Smith[3,4,17], V. Georgios Gkoutos [4,5,6], Krishnarajah Nirantharakumar [6,7], Zi Chong Kuo[8], Danni Wang[1,2], Qi Feng[9], Eddie C. Cheung[8,10], Lunzhi Dai [11], Junjie Huang [12], Yuanyuan Yu[13], Wenbo Meng [14] ✉, Xiwen Qin [15,16] ✉ & Jinqiu Yuan [1,2,8] ✉

Proton pump inhibitors (PPIs) are commonly used for gastric acid-related disorders, but their safety profile and risk stratification for high-burden diseases need further investigation. Analyzing over 2 million participants from five prospective cohorts from the US, the UK, and China, we found that PPI use correlated with increased risk of 15 leading global diseases, such as ischemic heart disease, diabetes, respiratory infections, and chronic kidney disease. These associations showed dose-response relationships and consistency across different PPI types. PPI-related absolute risks increased with baseline risks, with approximately 82% of cases occurring in those at the upper 40% of the baseline predicted risk, and only 11.5% of cases occurring in individuals at the lower 50% of the baseline risk. While statistical association does not necessarily imply causation, its potential safety concerns suggest that personalized use of PPIs through risk stratification might guide appropriate decision-making for patients, clinicians, and the public.

Proton pump inhibitors (PPIs) are currently the first-line treatment for acid-related disorders such as gastroesophageal reflux disease. PPIs are one of the top ten most prescribed drugs worldwide[1]. The global PPI market was evaluated at $2.9 billion in 2020 and is anticipated to grow by 6.9% annually from 2023 to 2030[2].

Although the pharmaceutical industry has made remarkable progress in novel drug discovery for safer and better PPIs, there has been a growing concern over the potential adverse effects of long-term PPI usage. In 2021, an umbrella review of 42 meta-analyses showed that long-term use of PPIs was associated with a variety of diseases, such as

[1]Department of Epidemiology and Biostatistics, Clinical Big Data Research Center, The Seventh Affiliated Hospital, Sun Yat-sen University, Shenzhen, Guangdong, China. [2]Chinese Health RIsk MAnagement Collaboration (CHRIMAC), Shenzhen, Guangdong, China. [3]Institute of Inflammation and Ageing, University of Birmingham, Birmingham, UK. [4]University Hospitals Birmingham NHS Foundation Trust, Birmingham, UK. [5]Institute of Cancer and Genomic Sciences, University of Birmingham, Birmingham, UK. [6]College of Medical and Dental Sciences, Centre for Health Data Science, University of Birmingham, Edgbaston, Birmingham, UK. [7]Institute of Applied Health Research, University of Birmingham, Birmingham, UK. [8]Guangdong Provincial Key Laboratory of Gastroenterology, Center for Digestive Disease, The Seventh Affiliated Hospital, Sun Yat-sen University, Shenzhen, Guangdong, China. [9]Oxford Population Health, University of Oxford, Oxford, Oxfordshire, UK. [10]Division of Gastroenterology, Davis School of Medicine, University of California, Oakland, CA, USA. [11]National Clinical Research Center for Geriatrics and Department of General Practice, West China Hospital, Sichuan University, Chengdu, China. [12]J.C. School of Public Health and Primary Care, Faculty of Medicine, The Chinese University of Hong Kong, Sha Tin, Hong Kong, China. [13]Department of Surgery, The Chinese University of Hong Kong, Sha Tin, Hong Kong, China. [14]Department of General Surgery, The First Hospital of Lanzhou University, Lanzhou, Gansu, China. [15]School of Population and Global Health, Faculty of Medicine, Density and Health Sciences, University of Western Australia, Perth, AU-WA, Australia. [16]Laboratory of Data Discovery for Health (D24H), Hong Kong Science Technology Park, Sha Tin, Hong Kong, China. [17]These authors contributed equally: Bin Xia, Qiangsheng He, Fang Gao Smith. ✉e-mail: mengwb@lzu.edu.cn; simon.qin@uwa.edu.au; yuanjq5@mail.sysu.edu.cn

chronic kidney disease (CKD) and enteric infection[3]. Moreover, the reporting of new PPI-associated adverse effects continues. In the past 2 years alone, epidemiological studies, for the first time, linked PPI use to increased risk of biliary tract cancer[4,5], rheumatoid arthritis[6], type-2 diabetes[7,8], inflammatory bowel disease[7], and cholelithiasis[8]. The evidence for these associations was generally observational. Reporting of these adverse events attracted increasing attention to the appropriate use of PPIs, while it also resulted in fears and reduced adherence to PPI treatment in patients. The need for personalized strategies for reducing unnecessary PPI use has become an urgent subject to be addressed[9,10].

There are several weaknesses in existing studies. First, most of the previous studies only evaluated single outcomes within one population. Some studies might be limited by selective reporting or lack of validation in different populations. Second, a comprehensive evaluation of the overall safety profile and dose–response effects, particularly for major diseases with top global disease burden remains lacking. Finally, and more importantly, there exists a major knowledge gap on which group of people are more vulnerable to PPI-related adverse events. Thus, individualized treatment based on patients' underlying risk may confer benefits and reduce harms. Such a risk stratification approach, successfully implemented in selecting patients for anti-hypertensive and statin therapy[11,12], has also been applied to individuate avoidance of additional risks related to PPI use, such as type-2 diabetes[7], stroke[13], and cholelithiasis[8]. However, its application for

other PPI-associated adverse events remains unclear. To address these concerns and improve decision-making for the appropriate use of PPIs in the public, patients, clinicians, and industry, we comprehensively evaluated the overall safety profile, dose–response relationships, and individual risk stratification for PPI use and the top 30 causes of global disease burden based on five large cohorts from the US, the UK, and China.

## Results

A total of 2,079,724 participants from UK Biobank ($n = 501,109$), Nurses' Health Study (NHS, $n = 91,708$), NHS II ($n = 99,641$), Health Professionals Follow-Up Study (HPFS, $n = 30,933$), and Clinical Data Analysis and Reporting System (CDARS, $n = 1,356,333$) were included as the basic population for the current analyses. (Fig. 1). The characteristics of included participants by cohort were summarized in Table 1 and presented in details by PPI use in Supplementary Tables S1–5. Overall, the participants had a mean age between 48.4 and 71.4 at baseline, and with a PPI usage rate from 5 to 15%. Compared with non-PPI users, regular PPI users were more likely to be older, obese, smoking, less physically active, with higher rates of comorbidities and medication usage.

Figure 2 presents the combined associations of PPI use and risk of the 30 leading causes of global disease burden (see details in Supplementary Tables S6–35). For a follow-up time of between 4 and 17 years, PPI use was positively associated with 15 of the 30 leading causes: (1)

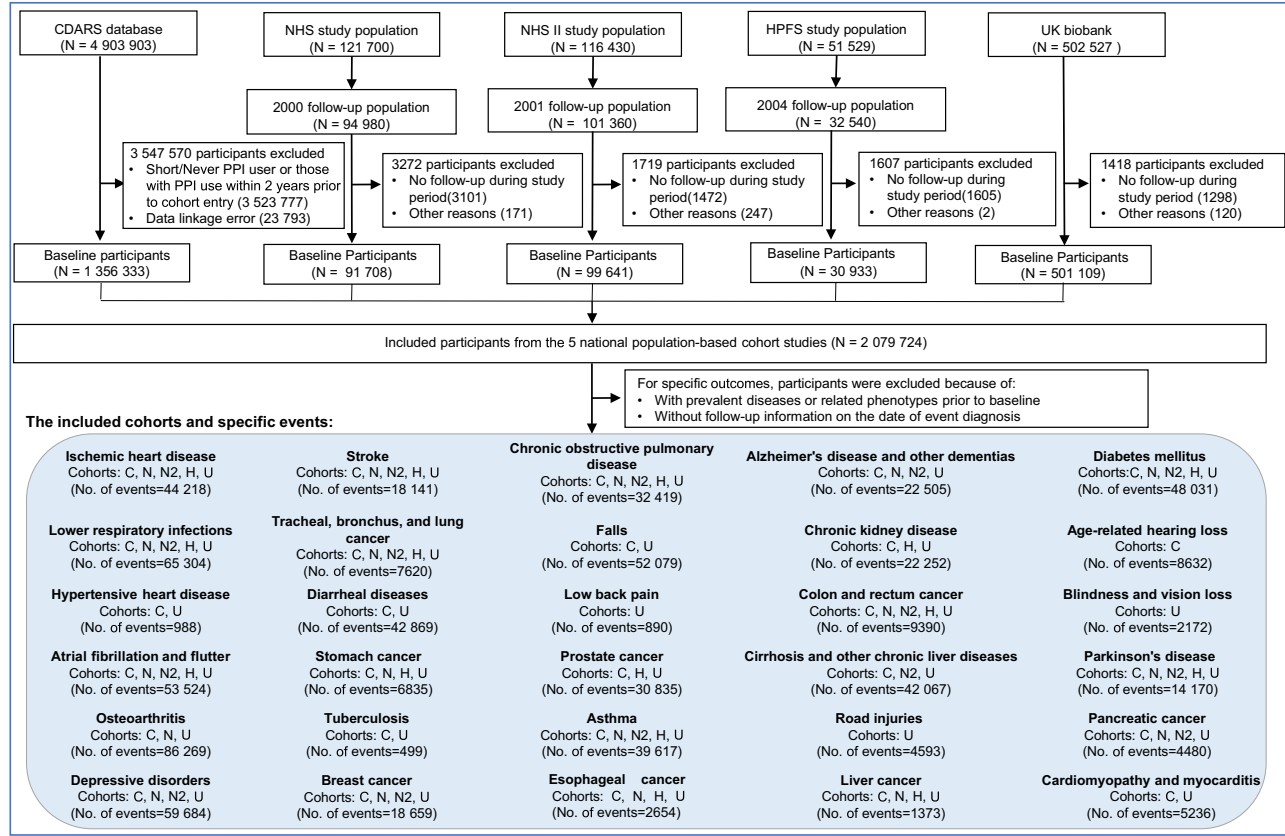

**Fig. 1 | Flowchart of participant inclusion.** This prospective analysis encompasses five population-based cohort studies: Clinical Data Analysis and Reporting System (CDARS), China; Nurses' Health Study (NHS), USA; Nurses' Health StudyII (NHS II), USA; Health Professionals Follow-Up Study (HPFS), USA; and UK Biobank, UK. We extracted data from 4,903,903 individuals from the CDARS who had information on the prescription of either a Proton pump inhibitor (PPI) or H2RA from January 1, 2003, through December 31, 2017. Exclusion criteria included individuals classified as short-term or non-users of PPI, those who used PPI within two years prior to cohort entry, and instances of data linkage errors. After exclusions, the CDARS baseline cohort consisted of 1,356,333 individuals. Similar exclusions (e.g., lack of follow-up or other reasons) were applied to the other cohorts, resulting in 91,708 individuals for NHS, 99,641 individuals for NHS II, 30,933 individuals for HPFS, and 501,109 individuals for UK Biobank. The final baseline study population included 2,079,724 participants across all five studies. For analysis of 30 leading causes of global disease burden, this study included cohorts with corresponding disease data, with the exclusion of participants with prevalent diseases or related phenotypes prior to baseline and availability of follow-up information on the date of event diagnosis for subsequent analysis.

## Table 1 | Baseline participant characteristics

| Variable | UK biobank (n = 501,109) | NHS (n = 91,708) | NHS II (n = 99,641) | HPFS (n = 30,933) | CDARS (n = 1,356,333) |
|---|---|---|---|---|---|
| Mean (SD) age (years) | 57.0 (8.1) | 68.4 (7.1) | 48.8 (4.6) | 71.4 (8.4) | 67.5 (7.0) |
| Female, no. (%) | 272,632 (54.4) | 91,708 (100.0) | 99,641 (100.0) | 0 (0.0) | 464,727 (34.3) |
| White, no. (%) | 474,203 (94.6) | 89,158 (97.2) | 95,721 (96.1) | 28,243 (91.3) | NA |
| Postmenopausal women, no. (%) | 194,109 (38.7) | 91,307 (99.6) | 49,182 (49.4) | 0 (0.0) | NA |
| Mean (SD) BMI, kg/m$^2$ | 27.4 (4.8) | 26.9 (5.4) | 27 (6.4) | 23.5 (8.8) | NA |
| Never smoker, no. (%) | 275,748 (55.0) | 42,126 (45.9) | 64,889 (65.1) | 14,507 (46.9) | NA |
| No-alcohol drinkers, no. (%) | 40,510 (8.1) | 33,296 (36.3) | 25,677 (25.8) | 5827 (18.8) | NA |
| Median (IQR) physical activity, MET hours/week | 29.5 (46.0) | 12.9 (30.3) | 16.7 (40.5) | 38.4 (69.9) | NA |
| >5 portions of fruit and vegetable per day, no. (%) | 189,346 (37.8) | 19,743 (21.5) | 17,600 (17.7) | 6398 (20.7) | NA |
| Mean (SD) AHEI score | NA | 50.8 (10.4) | 51.5 (10.3) | 56.1 (10.0) | NA |
| **Prevalent comorbidities, no. (%)** | | | | | |
| GERD | 36,188 (7.2) | 27,180 (29.6) | 27,648 (27.8) | 8721 (28.2) | 377,080 (27.8) |
| Gastric or duodenal ulcer | 11,825 (2.4) | 2819 (3.1) | 2441 (2.4) | 1205 (3.9) | 48,828 (3.1) |
| Upper gastrointestinal tract bleeding | 4095 (0.8) | 1718 (1.9) | 595 (0.6) | 737 (2.4) | 28,482 (2.1) |
| Cancer | 28,690 (5.7) | 13,675 (14.9) | 2871 (2.9) | 4234 (13.7) | 223,746 (16.5) |
| Hypertension | 294,329 (58.7) | 45,044 (49.1) | 17,471 (17.5) | 15,673 (50.7) | 694,442 (51.2) |
| Hypercholesterolemia | 93,596 (18.7) | 55,781 (60.8) | 29,409 (29.5) | 18,745 (60.6) | 293,893 (21.7) |
| Diabetes | 29,967 (6.0) | 8257 (9.0) | 3040 (3.1) | 3155 (10.2) | 453,348 (33.4) |
| **Current medication use, no. (%)** | | | | | |
| Multivitamin | 75,507 (15.1) | 52,393 (57.1) | 48,938 (49.1) | 19,408 (62.7) | NA |
| PPI | 51,241 (10.2) | 5526 (6.0) | 5222 (5.2) | 4550 (14.7) | 722,835 (53.3) |
| Aspirin | 71,928 (14.4) | 16,026 (17.5) | 9036 (9.1) | 15,837 (51.2) | 737,025 (54.3) |
| NSAIDs | 155,648 (31.1) | 59,496 (64.9) | 50,040 (50.2) | 19,390 (62.7) | 802,202 (59.1) |
| Statin | 81,728 (16.3) | 19,088 (20.8) | 6499 (6.5) | 11,295 (36.5) | 348,707 (25.7) |
| ACEIs | 49,453 (9.9) | 10,699 (11.7) | 4147 (4.2) | 4767 (15.4) | 230,576 (17.0) |
| Beta-blockers | 32,399 (6.5) | 14,435 (15.7) | 5802 (5.8) | 5078 (16.4) | 97,655 (7.2) |
| Calcium-channel blockers | 31,325 (6.3) | 7908 (8.6) | 2036 (2.0) | 2558 (8.3) | 112,575 (8.3) |
| Thiazide diuretics | 31,105 (6.2) | 12,179 (13.3) | 5267 (5.3) | 3168 (10.2) | 12,991 (9.6) |
| Antidepressants | NA | 9075 (9.9) | 15,556 (15.6) | 482 (1.6) | 31,722 (2.3) |
| Antibiotics | NA | 75,592 (82.4) | 85,244 (85.6) | 19,395 (62.7) | 19,786 (1.5) |

*SD* standard deviation, *BMI* body mass index, *IQR* interquartile range, *MET* Metabolic equivalent of task, *AHEI* Alternate Healthy Eating Index, *GERD* gastroesophageal reflux disease, *NSAIDs* non-steroidal anti-inflammatory drugs, *ACEIs* angiotensin-converting enzyme inhibitors.

ischemic heart disease (IHD), (2) stroke, (3) chronic obstructive pulmonary disease (COPD), (4) diabetes mellitus, (5) lower respiratory infections, (6) falls, (7) CKD, (8) diarrheal diseases, (9) atrial fibrillation and flutter, (10) cirrhosis and other chronic liver diseases, (11) Parkinson's disease, (12) osteoarthritis, (13) asthma, (14) depressive disorders, and (15) esophageal cancer. (HRs: 1.12–1.54). The *P* values were generally <0.01, except for osteoarthritis (0.013) and esophageal cancer (0.039). The results obtained through the H2RA active comparator design in the CDARS database align closely with the associations found in other cohorts for almost all outcomes associated with PPI use. Road injuries, which could be considered as negative control outcome, showed no association with PPI use as expected (HR = 1.02, 95% CI 0.91–1.14). In the evaluation of dose–response relationships in CDARS, the aforementioned 15 outcomes consistently showed an increased risk with the accumulated duration of PPI use (Fig. 3).

The primary results did not reveal major changes in the sensitivity analyses by lagging the exposure for 4 years, using propensity score analysis, and excluding CDARS (Supplementary Fig. S1). The *E* values ranged from 1.49 to 2.45 for the primary estimates (Supplementary Fig. S2).

Based on the UK Biobank and CDARS datasets, we found that there were generally no major differences in the disease risks among omeprazole, lansoprazole, esomeprazole, and other PPIs (Supplementary Fig. S3). In the subgroup analyses, most estimates for PPI-related risk did not differ by age, sex or BMI, except for diabetes mellitus, CKD, falls, or Parkinson's disease (*P*-interaction <0.05) (Supplementary Figs. S4–6).

We investigated risk stratification for the 15 identified diseases that were associated with PPI use in the primary results. This analysis was carried out in the UK Biobank since it collected the most comprehensive baseline and outcome data. The prediction models for evaluating baseline risk were described in Supplementary Data 1. For individual outcomes, we found that the RDs were consistently increased with the baseline predicted risk by ~3–26 times from quartile 1 to quartile 4 (Fig. 4), suggesting prevention of PPI-related risk should focus on the participants with high baseline risk.

For easier screening of high-risk individuals during clinical practice, we developed a prediction model that incorporates several key variables as predictors for a composite outcome of any of the 15 diseases. These predictors include age, BMI, the number of treatments or medications taken, smoking status, long-standing illness, overall health rating, and self-reported usual walking pace. Detailed information about these variables in the prediction model was presented in Table 2 (see Supplementary Fig. S7 for the nomogram of the prediction

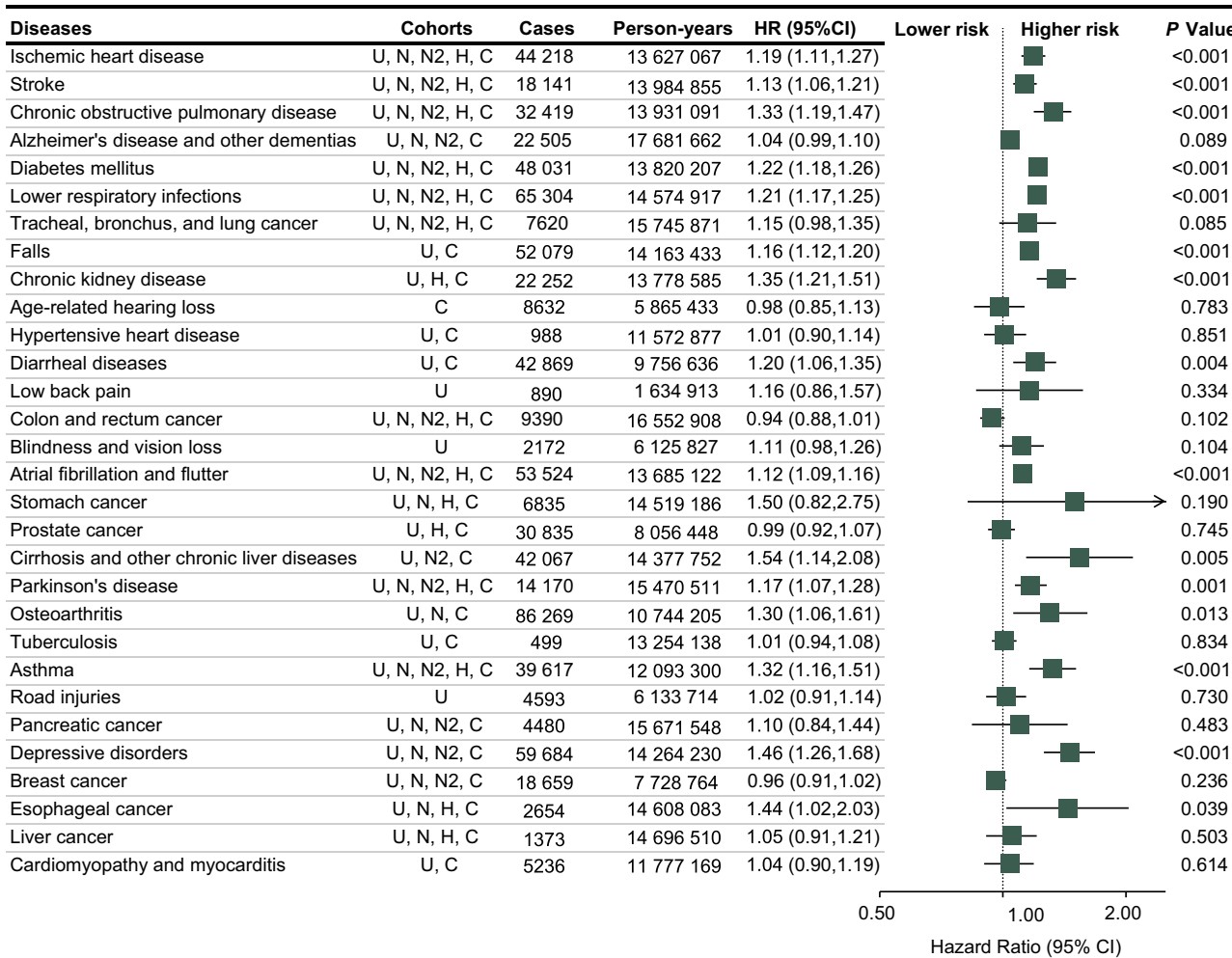

**Fig. 2 | Combined analyses of PPI use and risk of the 30 leading causes of global disease burden.** The squares denote the adjusted hazard ratio (HR), and the horizontal lines represent the 95% confidence intervals (CI). Two-sided *P* values were derived using inverse variance-weighted, random-effect meta-analyses (*N* = 2,079,724) and are not corrected for multiple testing. U UK Biobank, N NHS, N2 NHS II, H HPFS, C CDARS.

model). Although the performance of the model is moderate to low (c-index = 0.68. Supplementary Table S36 and Supplementary Fig. S8), the model could effectively stratify the PPI-related adverse events, with a risk difference (RD) of 2.94% for the individuals at the upper 20% predicted risk, compared with 1.07% for those at lower 20% (Fig. 5). We estimated that 82.1% of the annual number of cases that were attributed to PPI use occurred in the individuals at the upper 40% of the baseline predicted risk, and 42.3% of the cases occurred in those at the upper 10% baseline risk. In the individuals at the lower 50% of the baseline risk, the cases that were attributed to PPI use only took up 10.9%. In the sensitivity analysis excluding osteoarthritis, which take up 41.2% of the cases, we observed similar results (Supplementary Fig. S9).

## Discussion

Based on five cohorts, including over 2 million participants, the present study indicated that PPI use was associated with half of the top 30 diseases of global disease burden, with most of them exhibiting a dose–response relationship. PPI-associated net risk was stratified with the baseline predicted risk, which suggests that prevention of potential PPI-associated should be individualized.

Many previous studies have evaluated PPI-related long-term risk. We systematically searched PubMed, EMBASE, and Web of Science from the inception to December 1, 2022. We included the most updated meta-analyses and recent original studies. Our results are consistent with previous meta-analyses/original studies which showed a positive association between PPI and stroke[13], lower respiratory infections[14], falls[15], CKD[16], diarrheal diseases[17], asthma[18,19], pancreatic cancer[20], Parkinson's disease[21,22], and depressive disorders[23,24] (Supplementary Data 2). Similar to the most recent meta-analyses, we also found no evidence of associations between PPI use and risk of Alzheimer's disease[25], colon and rectum cancer[26], prostate cancer[27], and breast cancer[27]. Previous meta-analyses or original studies suggested that the risk of tuberculosis[28], pancreatic cancer[29], liver cancer[27], visual and auditory impairments[30] were increased with PPI use. However, these associations were not confirmed by our study, potentially because our study employed a more comprehensive approach to control for potential confounders. Specifically, we addressed confounding through (1) extensive adjustment for a wide range of covariates, (2) and the use of an active comparator in the CDARS database. Study heterogeneity, reflecting variations in population characteristics, could also contribute to the disparities in findings. Two recent meta-analyses did not find sufficient evidence of associations between PPI use and risk of IHD[31] and diabetes mellitus[32], while in the current study, we found significant associations and clear dose–response relationships. We noted that the aforementioned meta-analysis did not include 4 recent cohort or case-control studies[33–36], which all demonstrated a significant association between PPI and risk of developing type-2 diabetes. A preliminary meta-analysis of all published results

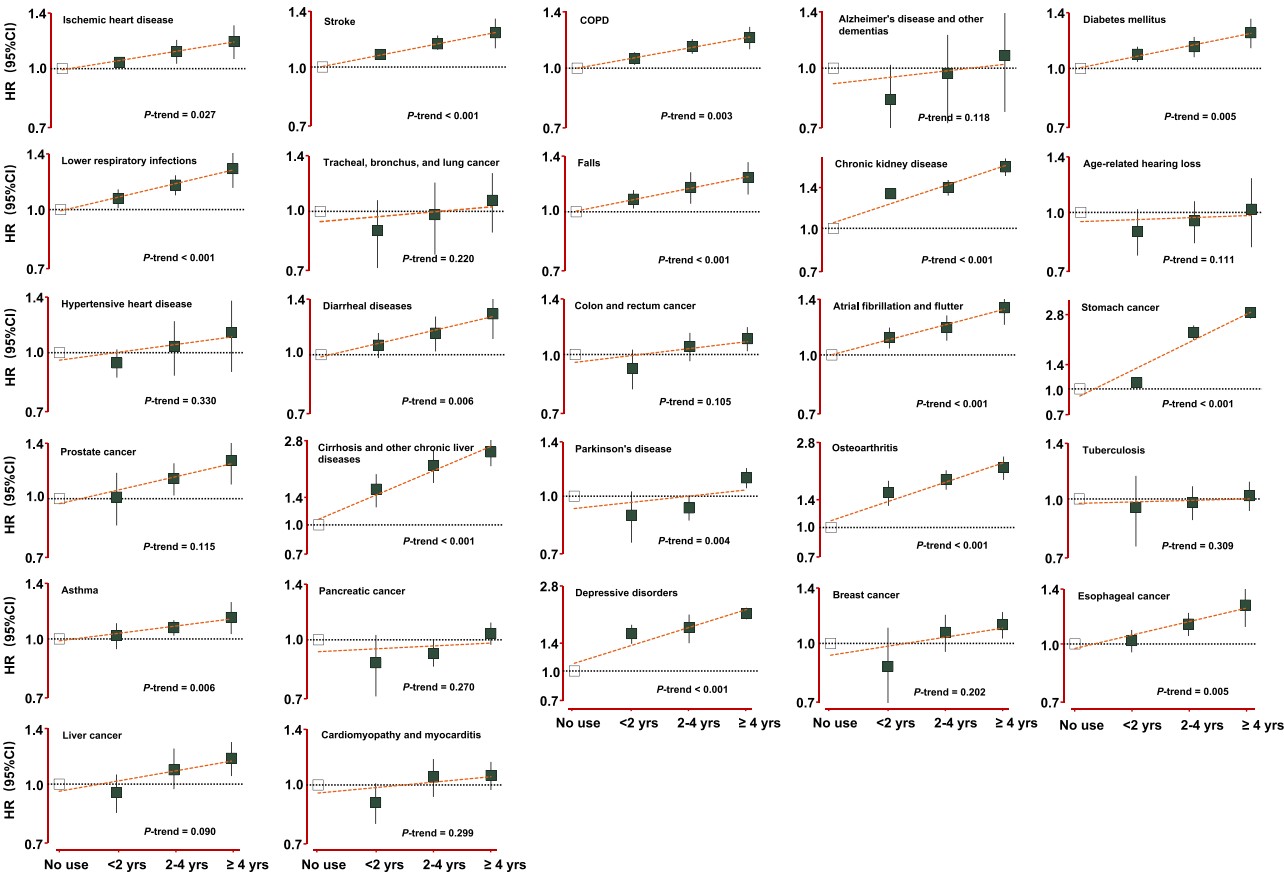

**Fig. 3 | Dose−response associations according to the accumulated duration of PPI use in CDARS.** The squares denote the adjusted hazard ratio (HR), and the vertical lines represent the 95% confidence intervals (CI). All displayed $P$ values are two-sided for trend analyses ($N = 1,356,333$) without adjustment for multiple comparisons. Source data are provided as a Source Data file.

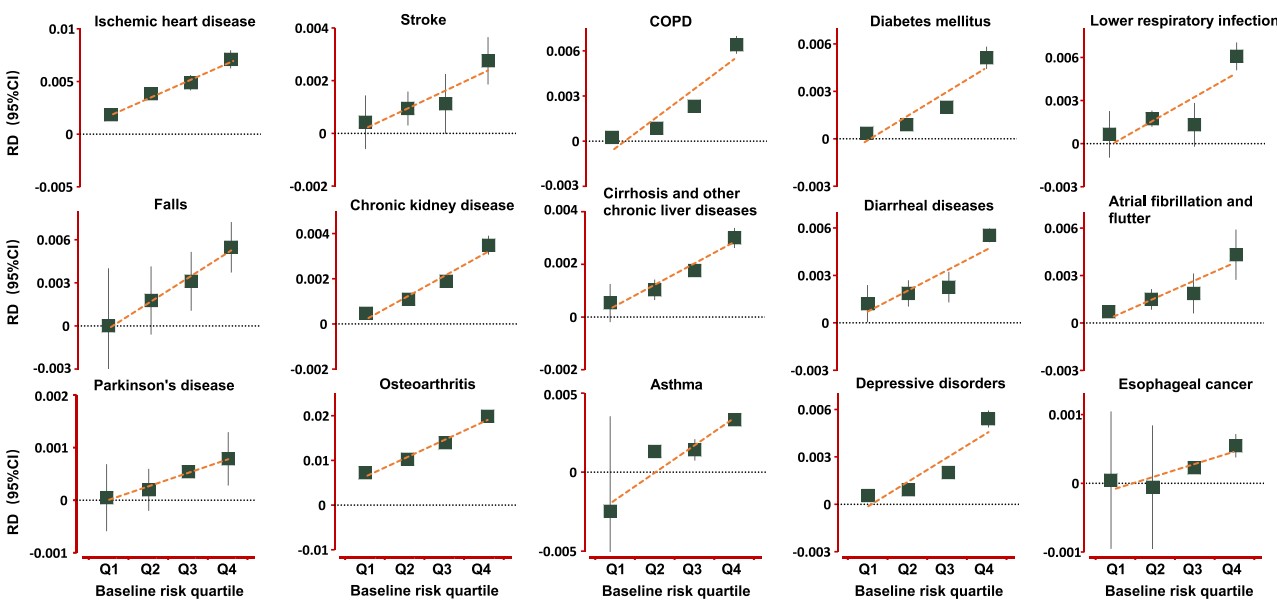

**Fig. 4 | Risk stratification for PPI-related adverse events.** The analysis was carried out for the 15 high-burden diseases that were associated with PPI use in the primary analysis based on the UK Biobank ($N = 501,109$). The baseline risk was evaluated with newly established models (lower respiratory infections, falls, diarrheal diseases, cirrhosis and other chronic liver diseases, asthma) or previously reported prediction models (Supplementary Data 1). We evaluated the HRs of PPI use as compared with no-PPI use in each quartile group, then translated to absolute effects (risk differences, RDs) associated with PPI use at one year by the method described by Altman[69]. The squares denote the RD, and the vertical lines represent the 95% confidence intervals (CI). Source data are provided as a Source Data file.

**Table 2 | Adjusted hazard ratios (95% confidence interval) for any of the 15 unintended outcomes in the derivation population**

| Variables | HR [95% CI] |
|---|---|
| Age, every 5 years increase | 1.35 [1.34, 1.35] |
| BMI, every 5 kg/m² increase | 1.16 [1.15, 1.17] |
| **Number of treatments or medications taken** | |
| 0 | 1.00 [Reference] |
| 1 | 1.08 [1.06, 1.11] |
| 2 | 1.18 [1.16, 1.21] |
| ≥3 | 1.34 [1.31, 1.36] |
| **Smoking status** | |
| Never | 1.00 [Reference] |
| Current | 1.60 [1.57, 1.64] |
| Previous | 1.15 [1.13, 1.17] |
| Long-standing illness | 1.30 [1.28, 1.32] |
| **Overall health rating** | |
| Poor | 1.00 [Reference] |
| Fair | 0.73 [0.70, 0.76] |
| Good | 0.54 [0.52, 0.56] |
| Excellent | 0.44 [0.42, 0.46] |
| **Self-reported usual walking pace** | |
| Steady average pace | 1.00 [Reference] |
| Slow pace | 1.23 [1.20, 1.27] |
| Brisk pace | 0.93 [0.92, 0.95] |

*BMI* body mass index, *HR* hazard ratio, *CI* confidence interval.

(HR = 1.17, 95% CI: 1.01–1.34, heterogeneity: $I^2 = 97\%$) suggested similar results as ours. Furthermore, our study failed to find a significant association between PPI use and gastric cancer as reported by the two recent meta-analysis[37,38]. While the direction of the pooled association in our study aligns with these prior reports, the lack of statistical significance may suggest that this large-scale study might still be underpowered for certain associations if a causal relationship does exist. This also raises the possibility that the situation may be even more challenging than the associations presented for the 15 important diseases.

However, statistical significance does not always indicate clinical relevance, especially with the relatively modest HRs ranging from 1.12 to 1.54 in our study. The PPI-associated absolute increase in risk, while statistically significant, should be interpreted cautiously in terms of clinical significance. In our prediction model that effectively stratifies the PPI-related adverse events, the absolute risk difference (RD) was 2.94% for individuals at the upper 20% of the baseline predicted risk, suggesting the degree of effect was small to modest in clinical practice.

The intricate mechanisms linking PPI use to a spectrum of morbid conditions are multifaceted. By blocking acid production, PPI impairs one of the body's natural defense mechanisms against ingested microorganisms, triggering profound changes in the gut microbiome[39]. This dysbiosis is evident in diarrheal diseases and involves the overgrowth of stomach bacteria, potentially increasing the risk of pneumonia through micro-aspiration[40]. The disturbance in the balance of microbial species in the gut and lungs may contribute to asthma through hyperactivation of T helper cell-dominated immune responses and the overproduction of inflammatory cytokines, leading to airway inflammation[41]. Moreover, disruptions in the microbiome facilitate bacteria producing nitrosamines, and bile salt toxicity due to elevated stomach pH are potential mechanisms for an increased risk of esophageal cancer[42]. Enterococcus growth in the intestines translocating into the liver and inducing inflammation may contribute to chronic

liver diseases[43]. Beyond the gastrointestinal realm, the derived hypomagnesemia and reduced insulin-like growth factor-1 (IGF-1) levels might facilitate diabetes mellitus development[44]. PPIs' impact on enteric infection, along with hypomagnesemia and uremic toxin accumulation, may contribute to CKD[45]. In neurological implications, small intestinal bacterial overgrowth (SIBO) and subsequent inflammatory responses are linked to Parkinson's disease[46]. PPI interference with the microbiome, hypergastrinemia, and potential impacts on central nervous system immune activity are suggested mechanisms for depressive disorders[47]. Beyond microbial effects, diminished gastric acidity in PPI users causes calcium or vitamin B12 malabsorption, decreasing bone mineral density and elevating the risk of osteoporosis and falls[48]. In addition, PPIs may affect cardiovascular risk by modulating plasma asymmetric dimethylarginine, reducing nitric oxide levels, and impairing endothelium-dependent vasodilation[49]. It is essential to note that these findings are primarily derived from ex vivo studies, necessitating further investigation to elucidate the intricate associations observed.

The identified associations may be partially or completely due to confounding effects. Potential confounders include indications for using PPIs, overall health status, comorbidity, and other medications. To minimize confounding effects, our study first comprehensively adjusted for potential confounders in statistical models, then included propensity score analysis, and finally, for the CDARS cohort, contributing to over 65% of the overall population, H2RAs were considered as active comparators. Some of the positive associations, such as osteoarthritis and esophageal cancer, may be due to reverse causation, whereby the subclinical symptoms may be related to PPI use. In the primary analysis, we lagged the exposure for a time window of 2 years (4 years in the sensitivity analysis) to minimize its influence. Randomized clinical trials (RCTs) and RCT-based meta-analyses would provide the best evidence on PPI safety, which is currently unavailable (except for enteric infection[50] and stroke[8]) and may not be ethically or practically feasible in the foreseeable future.

Clinicians now face a dilemma: there is a lacking of high-quality evidence confirming the long-term safety of PPI use, and their potentially harmful effects cannot be ignored, since should these associations be causal, their impacts would be substantial. Our risk stratification approach provides a feasible practical solution. Risk stratification, in this context, facilitates identifying individuals at higher or lower baseline risk for PPI-associated adverse events. This involves a comprehensive assessment of individual characteristics and health status through a prediction model. The importance of risk stratification is not only to identify those who are at high risk and take preventive measures individually to minimize the additional harm caused by long-term PPI use, but also to screen patients who could safely use PPIs. This, in turn, reduces fears and increases treatment adherence among patients.

To the best of our knowledge, this is currently the most comprehensive assessment on the long-term safety of PPIs, encompassing the associations with 30 leading causes of global disease burden. The outcome-wide approach allowed us to compare the effects of multiple outcomes, which reduced selective reporting bias. In addition, disease incidence information was ascertained by national record linkage or biennially updated information, which might reduce misclassification, recall bias, and attrition bias. Finally, the dose–response associations, robust sensitivity analyses as well as the negative control outcome (road injuries), added additional strengths to our findings.

This study has its limitations. First, owing to the nature of the observational study, we could not ratify the causal relationship. This is particularly noteworthy as regular PPI users, in comparison to non-users, were more likely to be older, obese, smokers, less physically active, and had higher rates of comorbidities and medication usage. Although we made efforts to control for various confounders, residual confounding remains a possibility. Second, the findings might

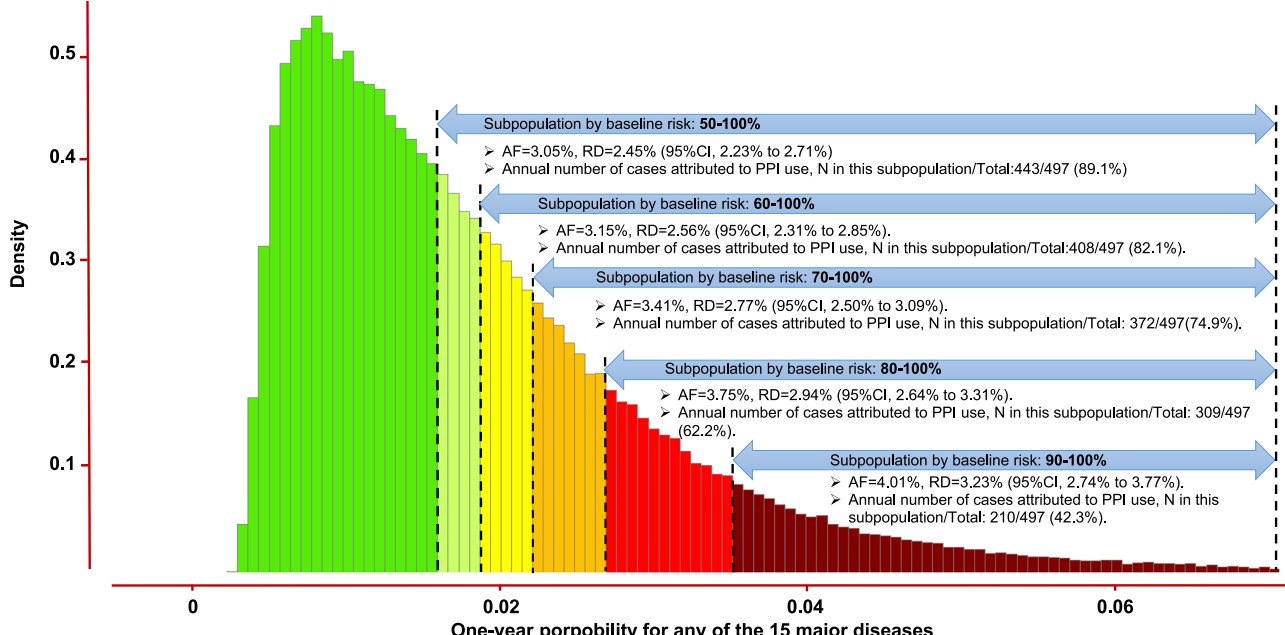

**Fig. 5 | Absolute risk of PPI-related high-burden diseases according to the distribution of the baseline predicted risk.** Abbreviation: AF, attributable fraction; PPI, proton pump inhibitor; RD, risk difference. This histogram presented the distribution of baseline predicted risk for any of the 15 PPI-related diseases. The performance and nomogram for the prediction model is available in Supplementary Table S36 and Supplementary Figs. S7 and 8. The RD and AF of PPI use for 1 year was calculated. Annual number of cases attributed to PPI use in each strata (i.e., *n*) was calculated based on attributable risk and exposure, and then summed as the total annual number of cases attributed to PPI use in all populations (i.e., total). The results showed that most cases were occurred in the individuals with high baseline predicted risk, and those who with low baseline risk do not need to be over panicked and should adhere to PPI treatment. Source data are provided as a Source Data file.

not be generalizable to all adult populations because the NHS, NHS II and HPFS included only health professionals, and a "healthy volunteer" effect may be prevalent within the UK Biobank. Third, the definition of PPI use and endpoints is not always consistent among the included cohorts. Furthermore, PPI use was only evaluated once at baseline in the UK Biobank, introducing a chance of misclassification during follow-up. Misclassification could underestimate the true effects, as the control group may include individuals who initiated PPI use during the follow-up period. To minimize the potential influence, we combined the effects with random-effect model as in other studies[7,8]. Fourth, some data were back-dated 10–20 years and may not accurately reflect the newly developed PPIs. Fifth, despite our study's large sample size and extensive range of outcomes, it may still be underpowered for certain associations, such as the one between PPI use and gastric cancer. In addition, as not all outcomes were available in every cohort, analyses for "Low back pain" and "Blindness and vision loss" still rely on data from a single cohort, which limits our ability to assess dose–response relationships for PPIs and conduct comparisons with a positive control. Further research with larger sample sizes is needed to strengthen the evidence regarding the potential effects of PPIs on these specific outcomes. Lastly, the results might be influenced by the immortal time bias. Nevertheless, the influence would be minor since we adopted an active comparator study design in CDARS, and used time-varying analyses in NHS, NHS II and HPFS, where the person-time at risk is defined after the initiation of treatment.

Given the links with a substantial global disease burden and the high rate of inappropriate overuse of PPIs (up to 70%)[51], the potential impact of long-term PPI use should not be ignored, even if the causal effects for these outcomes have not been established. In practice, the net risk of PPI-related adverse effects is low in those with low baseline risk, but it is not negligible and the risk profile may change over time.

Personalized prevention is feasible by regular evaluating the baseline risk with readily available predictive factors for long-term PPI users while focusing on the high-risk patients. For high-risk individuals, potential effective strategies, such as dose reduction, discontinuation, transitioning to "on-demand" use, considering less profound acid suppressants like H2RAs, and regular monitoring for early indications of adverse events (e.g., blood glucose levels for the risk of diabetes[9,10]), may help mitigate the additional absolute risk associated with PPI use.

Further research is still required to (1) confirm the causal effects of PPIs on disease risk through RCT-based meta-analysis; (2) improve and validate the performance of prediction models for multiple PPI-related adverse effects; (3) investigate the appropriate cut-off value for defining high-risk population; (4) evaluate the effectiveness of the risk stratification strategy. These research avenues have the potential to refine clinical practices and optimize PPI use, ensuring a balanced approach between therapeutic benefits and potential risks in diverse patient populations.

In conclusion, long-term use of PPIs is likely to be associated with a wide range of high-burden diseases. Although it remains unclear whether these associations are causal, its impacts should not be ignored due to potential links with heavy disease burden. The risk stratification approach by individualized using of PPIs after evaluating the PPI-related risk, may be an effective strategy to reduce potential risks as well as fears among patients.

## Methods
### Study design
This is a prospective analysis of five population-based prospective cohort studies including the UK Biobank (UK), NHS (USA), the NHS II (USA), HPFS (USA), and CDARS (China). The ethical committees from these cohorts approved the study. These cohorts have been utilized to evaluate PPI safety in many previous studies[6–8,13,52]. Study design of the present study was shown in Supplementary Fig. S10. In the current

analysis, we included participants who reported information about PPI usage and excluded those with outcomes of interest prior to baseline (see detailed information of participant inclusion in Supplementary Figs. S11–40). For the UK biobank, NHS, NHS II, and HPFS, we considered the non-regular PPI users as control. For CDARS, we used an active comparator design, taking H2 receptor antagonists (H2RAs), a class of less profound acid suppressants, as the control. This helps mitigate potential bias related to the clinical indications for which PPIs are prescribed. We performed an outcome-wide approach to identify diseases associated with PPI use in each cohort, and then pooled the associations with meta-analyses to get the overall estimates, which may lead to more conservative results.

The UK Biobank is a prospective cohort study of over 0.5 million adults recruited throughout England, Wales, and Scotland between 2006 and 2010. Participants aged 37–73 years with valid baseline data were followed up ever since for morbidities and mortality. Data in this study were obtained from UK Biobank (application number 51671, approved August 2019). Detailed description of the study design and survey methods of the UK Biobank cohort is available online (https://www.ukbiobank.ac.uk/media/gnkeyh2q/study-rationale.pdf). The UK Biobank was approved by the National Research Ethics Committee (REC ID: 16/NW/0274). Electronic written informed consent was obtained from all participants. The Nurses' Health Studies (NHS) comprise female registered nurses in the U.S. In 1976, 121,700 women between 30 and 55 years of age were included in the NHS cohort. In 1989, 116,430 female registered nurses between 25 and 42 years of age were enrolled in NHS II. The Health Professionals Follow-up Study (HPFS) comprises 51,529 US male dentists, pharmacists, veterinarians, optometrists, osteopathic physicians, and podiatrists, aged 40–75 years in 1986. All individuals completed a baseline mailed questionnaire on their medical history and lifestyle characteristics. Every other year, follow-up questionnaires are sent to these 3 cohorts to collect and update information on lifestyle factors and newly diagnosed medical conditions. The response rates have consistently exceeded 90%. The recruitment and data collection in the three cohorts have been reported in detail elsewhere[53–55]. In the current analysis, we included participants who reported information about PPI usage in 2000 for NHS, 2001 for NHS II and 2004 for HPFS. The NHS, NHS II were approved by the Human Research Committee at the Brigham and Women's Hospital, and the HPFS was approved by the Human Subjects Committee by the Harvard T H Chan School of Public Health, Boston, Massachusetts, USA. Clinical Data Analysis and Reporting System (CDARS) is an electronic database managed by the Hong Kong Hospital Authority. It is the sole public healthcare provider for primary, secondary and tertiary health services through seven hospital clusters and covers 87%–94% of all secondary and tertiary care in Hong Kong with a population of around 7.3 million[56]. This database was established in 1995 for both audit and research purposes. In CDARS, each participant was assigned a unique and anonymous patient identifier with linkage to the electronic Clinical Management System (e-CMS), recording routine clinical information including demographics, hospitalization, diagnoses, laboratory results, medication dispensing and death. For this study, cohort entry was defined as the date of this first prescription of either a PPI or H2RA from January 1, 2003, through December 31, 2017, and the follow-up duration of individual patient was defined as the duration of observation between the entry date and the censored date (December 31, 2020). To be included in the cohort, patients were required to have at least 2 year of medical information in the CDARS before cohort entry which served as a washout period to ensure new use of PPIs and H2RAs. This study protocol was approved by the Hong Kong Hospital Authority. This study was exempted from consent as all the data have been anonymised, and none of the authors were involved in data collection.

## Exposure assessment

In the UK Biobank, information on PPI use and class (including lansoprazole, omeprazole, pantoprazole, esomeprazole, and rabeprazole) was recorded by participants using a touchscreen questionnaire, and it was then verified during interviews conducted by research nurses. "Regular use" of medications was defined as "most days of the week for the last 4 weeks"[8]. In the case of the NHS, NHS II, and HPFS, participants were asked, in each biennial questionnaire, whether they had regularly, defined as "2+ times/week", used PPIs during the past 2 years. In CDARS, the detailed PPI prescription information, including drug name, and duration, were recorded. PPI users were defined as those who used PPIs for >30 days within the 2-year-cohort entry period. To eliminate the residual effects of previously used PPIs, we excluded the participants who had used any PPIs 2 years before cohort entry. We calculated the overall duration of PPI use by summing up each prescription period (days) recorded in the CDARS. The detailed questions regarding PPI use for these cohorts were reported elsewhere[52,57,58].

## Outcome ascertainment

The outcomes of interest were the top 35 causes of global disease burden in adults, aged 50 years and above, according to the Global Burden of Disease Study[59]. The scope was determined by considering and reviewing similar outcome-wide epidemiological studies. We excluded five causes (e.g., oral disorders, urinary diseases and male infertility, other musculoskeletal disorders, and endocrine, metabolic, blood and immune disorders) since these were not well-defined and/or were likely to be indications of PPIs (i.e., upper digestive system diseases), resulting in 30 diseases that were included in our final analysis. We used the International Classification of Diseases, Tenth Revision (ICD-10) or ICD-9 codes for medical condition identification in the UK biobank and CDARS through linkage to hospital inpatient records, as well as to cancer and death registries. In the NHS, NHS II and HPFS, cases assessment relies on self-report of clinical diagnosis or disease-specific medications on each biennial follow-up questionnaire. Supplementary questionnaires were further mailed to confirm the diagnoses, and related medical records were reviewed by study physicians. For conditions identified across multiple records of the same individual, the first record was used as the date of diagnosis. The accuracy and validity of the disease coding and diagnosis in these cohorts have been verified in previous studies[60–63].

## Assessment of covariates

In the baseline touchscreen questionnaire for the UK Biobank and each biennial questionnaire for the NHS, NHS II, and HPFS, participants provided personal information related age, ethnicity, body mass index (BMI), smoking, alcohol consumption, family history of diseases, multivitamin intake, comorbidities (e.g., gastroesophageal reflux disease, gastric or duodenal ulcer, gastrointestinal bleeding, hypertension, diabetes, and dyslipidemia), and medication usage (H2RAs, nonsteroidal anti-inflammatory drugs (NSAIDs), aspirin, and statins). Physical activity was assessed by the International Physical Activity Questionnaire (IPAQ) and diet intake was evaluated using the food frequency questionnaire, which has been validated in previous studies[64,65]. The UK Biobank also collected data on daily sleep duration, overall health rating, and the presence of a long-standing illness (Yes/No) at baseline. Overall diet quality (by 2010 Alternative Healthy Eating Index (AHEI-2010)) was only assessed in the NHS, NHS II, and HPFS. For the CDRAS dataset, we identified the aforementioned medications and comorbidities through the linkage of hospitals and pharmacy records.

## Statistics and reproducibility

**Statistical analyses.** No sample size calculation was conducted for this analysis, as it was based on secondary data from five international cohort studies and databses. Person-years were calculated from the

date of return of the baseline questionnaire (from 2006 to 2010 in the UK Biobank), first assessment of PPI use (2000 in the NHS; 2000 in the NHS II; 2004 in the HPFS), or cohort entry (from 2003 to 2017 in the CDARS) to the date of diagnosis of endpoint events, death, the end of follow-up, whichever occurred first. For CDARS, switching between PPIs to H2RAs during follow-up was also considered as a censoring indicator. We estimated multivariable-adjusted hazard ratios (HRs) and 95% confidence intervals (CIs) with Cox proportional-hazards models. To address potential reverse causation (i.e., symptoms of undiagnosed diseases resulting in PPI prescription), our analyses were restricted to patients with at least 2 years of follow-up after cohort entry, introducing a 2-year exposure-lag period. This approach aims to strengthen the temporality of our analysis by allowing for a sufficient latency period for disease risk development while also minimizing the impact of detection bias. We tested the proportional-hazards assumption by evaluating interactions between age and main exposures in time-varying Cox regression models in the NHS cohorts and HPFS, and by Schoenfeld tests in the UK Biobank and CDARS. No violation of this assumption was found. All analyses were performed using the SAS software, Version 9.4 (SAS Institute, Cary, North Carolina, USA) and R software (R Foundation for Statistical Computing, Vienna, Austria, version 3.5.0).

**Main analyses.** For the three US cohorts, we applied multivariable time-dependent Cox regression models stratified by age and time period (in 2-year intervals) and additionally adjusted for ethnicity, BMI, smoking status, alcohol consumption, physical activity, overall diet quality (AHEI-2010), portions of fruit and vegetable intake, family history of specific diseases, clinical indication for PPI use (i.e., GERD, gastric or duodenal ulcer, gastrointestinal bleeding), medications (multivitamin use, NSAID, aspirin, statin, ACEIs, beta-blockers, calcium-channel blockers, thiazide diuretics, metformin, antibiotic, oral steroids), comorbidities (diabetes, hypertension, hypercholesterolemia), and female-specific indicators (i.e., parity, menopausal status, and postmenopausal hormone use, for NHS and NHS II only). For the UK Biobank, we stratified the analyses jointly by age, sex, and UK assessment centers, adjusting for similar variables (i.e., demographic factors, lifestyle habits, medications, comorbidities, PPI clinical indications) as in the US cohorts, and additionally adjusted for self-reported overall health rating and long-standing illness. In the multivariable Cox regression models for the CDARS database, we stratified by age and sex, and additionally adjusted for the medications, comorbidities, and PPI indications mentioned above. We pooled the estimates of each cohort with inverse variance-weighted, random-effect meta-analyses[66]. Heterogeneity was evaluated with I² statistic[67]. To present the associations in a clinically translatable way, we calculated risk differences (RDs) based on the method described by Altman and Andersen[68]. For the UK biobank and CDARS cohorts, we repeated the main analysis by type of PPIs (omeprazole, lansoprazole, esomeprazole and other PPIs). To investigate potential dose–response relationship, we evaluated the associations between the cumulative duration of PPI use and disease risk in CDARS. To evaluate potential interaction effects with age, sex and body mass index (BMI), we undertook subgroup analyses and fitted an interaction term between PPI and these factors in the primary models.

**Sensitivity analyses.** To check the robustness of the primary results, we performed a number of sensitivity analyses. First, we lagged the exposure for an even longer time (4 years). Second, we carried out a propensity score (PS) analysis using the inverse probability treatment weighting (IPTW) method in the UK Biobank and CDARS database to adjust to potential bias in the allocation of patients to PPI. The propensity score was estimated using a logistic regression model which included all of the aforementioned covariates as potential predictors for PPIs. A weight was then calculated for each patient as 1/PS in the high-adherence group and 1/1-PS for those in the non-PPI user groups (UK Biobank) or H2RA user groups (CDARS database). Extreme weight values were truncated at the 5th and 95th percentile ends of the distribution. We confirmed that the IPTW method (through weighting) had adequately balanced the covariate profile of the two groups by comparison of the unweighted and weighted standardized difference in means/proportions for each covariate.Third, we pooled the estimates of each cohort after excluding the CDARS database. Last, we calculated the E values to evaluate the potential influence of unmeasured confounders[69].

**Risk stratification.** We tested risk stratification based on the UK biobank. For the outcomes that were associated with PPI use in the primary analyses, we reviewed the literatures for appropriate prediction models (Supplementary Data 1). If a prediction model was not available, we developed new prediction models using major known risk factors. Since all of the considered outcomes were undesired and separate models would largely increase the complexity of evaluation in practice, we established a prediction model for a composite outcome for any of the identified significant events. For ease use, we only included candidate predictors that are readily available, such as age, BMI, smoking, long-standing illness, self-rated overall health. We stratified the participants based on the predicted risk, and then, evaluated the absolute effect (by RD) of PPI use in each quartile risk group. We also perform sensitivity analysis by excluding osteoarthritis, which took up 41.2% of the cases, to test the robustness of the results.

**Derivation and validation of the prediction model for any of the 15 PPI-associated high-burden diseases.** We utilized Cox regression models to identify potential predictors associated with the occurrence of any of the 15 PPI-related diseases, aiming to construct a comprehensive predictive model and estimate coefficients for each identified risk factor. Initially, we created a composite indicator to signify the occurrence of any of the 15 PPI-associated high-burden diseases. The primary time variable considered in this analysis was the duration from UK Biobank entry until the first instance of any of the 15 PPI-related diseases, death, or loss to follow-up, whichever came first.

We examined a range of variables for potential inclusion in the prediction model, encompassing age at study entry (continuous), sex (male, female), body mass index (continuous), ethnicity (white, or other), education levels (less than high school, high school or equivalent, or college or above), smoking status (never smoked, previous smoker, current smoker), alcohol consumption (daily or almost daily, one to four times a week, one to three times a month, special occasions only or never), physical activity (low, moderate, or high), fruit and vegetable intake (≥5 portions or <5 portions), red and processed meat intake (<2.0 times per week, 2.0–2.9 times per week, 3.0–3.9 times per week, and ≥4.0 times per week), sleep time (<8 h, 8 h, 8–9 h, >9 h), prevention of hypertension, hyperlipidemia, overall health rating (poor, fair, good, excellent), long-standing illness (yes or no), and self-reported usual walking pace (slow, steady average, brisk pace). All these candidate predictors represent established factors influencing the incidence of any of the 15 outcomes and are typically obtained through questionnaires.

Variables were retained in the prediction model if they exhibited a hazard ratio of <0.85 or >1.15 (for binary variables) and achieved statistical significance at the 0.01 level. The prediction model was initially derived from participants in England and subsequently geographically validated in participants from Scotland and Wales. Discrimination of the prediction model was evaluated using C-index over 1, 5, and 10 years, while calibration was assessed using calibration plots based on risk deciles at 5 years. In addition, we constructed a nomogram for the model to offer a more direct means of assessing the 1-, 5-, and 10-year probability of any of the 15 unintended outcomes.

**Reporting summary**

Further information on research design is available in the Nature Portfolio Reporting Summary linked to this article.

## Data availability

The UK Biobank data used in this study was available from UK Biobank under application number 51671. The CDARS database is under restricted access to researchers upon approval by the Hong Kong Hospital Authority. NHS, NHS II and HPFS are available upon formal application to and approval by the Channing Division of Network Medicine at Brigham and Women's Hospital, and Harvard T.H. Chan School of Public Health. For the protection of confidentiality and privacy of cohort participants, the written request for access to the data is required. The standard procedure for controlled access requires that applications to use the resources of the Nurses' Health Studies and Health Professionals Follow-up Study undergo a formal review by the cohort committee. The committee assesses the scientific aims, examines the suitability of the proposed methodology for the available data, and confirms that the proposed use aligns with the guidelines of the Ethics and Governance Framework. Further information including the procedures to obtain and access data from the NHS, NHS II and HPFS is described at https://www.nurseshealthstudy.org/researchers (email: nhsaccess@channing.harvard.edu) and https://sites.sph.harvard.edu/hpfs/for-collaborators/. Requests for access to raw data from the study should be addressed to the corresponding authors and will be answered within 12 weeks. Source data are provided with this paper.

## Code availability

The analytic SAS and R code is available through a git repository at https://github.com/Jinqiu-Yuan/Personalized-use-of-PPIs.

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

## Acknowledgements

The authors would like to thank Prof. Jae Hee Kang, and Prof. Heather Eliassen (Brigham and Women's Hospital, Harvard Medical School, Boston, MA) for their help in sharing data and comments on data analysis. We also acknowledge the dedication of the participants and staff from UK Biobank, NHS, NHS II, HPFS, and CDRAS. This research was supported by the Natural Science Foundation of China [grant number 82103913 (to Q.S.H.), 82003524 (to J.Q.Y.), 82003408 (to B.X.)], Guangdong Provincial Key Laboratory of Digestive Cancer Research, No. 2021B1212040006 (to J.Q.Y.), the Start-up Fund for the 100 Top Talents Program, SYSU No.392012 (to J.Q.Y.), and the Research Supporting Start-up Fund for Associate researcher of SAHSYSU, Grant No. ZSQYRSSFAR0004 (to B.X.). The funders had no role in the design and

conduct of the study; collection, management, analysis, and interpretation of the data; preparation, review, or approval of the manuscript; or decision to submit the manuscript for publication.

## Author contributions

Prof. Jinqiu Yuan had full access to all of the data in the study and take responsibility for the integrity of the data and the accuracy of the data analysis. Bin Xia, Qiangsheng He, and Jinqiu Yuan were involved in the acquisition, analysis, or interpretation of data. Bin Xia, Qiangsheng He, and Fang Gao Simith drafted the manuscript. All authors critically reviewed the manuscript for important intellectual content. V. Georgios Gkoutos, Krish Nirantharakumar, Zi Chong Kuo, Danni Wang, and Qi Feng contributed to the intellectual content. Bin Xia, Qiangsheng He, and Jinqiu Yuan obtained funding for the study. Eddie C. Cheung, Lunzhi Dai, Junjie Huang, Yuanyuan Yu, Wenbo Meng provided administrative, technical, or material support. Wenbo Meng, Xiwen Qin, and Jinqiu Yuan supervised the study. All authors have read and approved the final version of the manuscript. Bin Xia, Qiangsheng He, and Fang Gao Simith contribute equally to this work.

## Competing interests

The authors declare no competing interests.
