## [Peer Review File · Nature Communications]

REVIEWER COMMENTS

Reviewer #1 (Remarks to the Author):

I read with interest the paper examining adverse events in PPI users using several large previously published cohorts. The authors included a novel approach to assess the multitude of outcomes associated with PPI use, and these results are surprisingly consistent among the different study cohorts, with dose-response curves supporting the proposed causal association for almost all outcomes. This article also includes the construction of a prediction model, to predict who is most at risk of any of these 15 outcomes (and for whom discontinuation should especially be considered?), yet I do wonder if there shouldn't be any efforts to discontinue treatment in all receiving inappropriate PPI use.

- The paper is interesting, yet there is a lot of information in there, with the estimation of effects prediction model and also some sort of evidence synthesis (which is not described in detail yet seems to have caught many/most of the relevant and recent papers). The authors also succeeded in properly describing the problem and potential the underlying mechanisms – and the methodology seems appropriate

- I think the part on the prediction model and how to interpret/use it, is a bit hard to follow (because of limited word count and many important parts moved to the supplements).

- Regarding clinical implications: Those who are not at high risk now, may be in a few years - and it seems almost all cases of these adverse events in the low risk groups are attributed to PPI use. In absolute numbers, these outcomes are indeed most frequent in the “high risk” but relatively speaking, the risks do not seem negligible in the low-risk groups? The authors did not specifically mention the inappropriate over-use of PPIs which is estimated to be up to 70% in a Cochrane review.

- I do hope this paper helps in the awareness among clinicians, as a broad range of outcomes should be considered when prescribing PPIs, ideally restricted to well-established indications. There has been a lot of discussion on the harm vs. benefit of PPIs, and the risks of individual outcomes – but the present paper does a good attempt to make a composite outcome – although this still seems likely/plausible to be an underestimation of the true effect (assuming causality) as some analysis seem underpowered?

- I think the article could use a discussion on the association with gastric cancer. Many studies have been published on this association, even a handful of meta-analyses (some mentioned in supplements). I was first not sure if it was not in that list of 30-35 diseases, or that it didn't come out as significant. Digging into the data, I found that it was the latter scenario – and it is remarkable that the pooled effect size of gastric cancer is lower than many other (significant) associations. This may suggest that this large study is still underpowered for some associations – and that it is “even worse” than the presented association with 15 important diseases.

- I would put Supplementary Table 37 and Suppl Fig 7 as main Table

- The prediction model is of great clinical importance as it could guide in whom discontinuation of PPI treatment should be most beneficial. I would at least mention which variables ended up in the prediction model (in the results section)

- I am not sure how the authors applied the H2RA active comparator design, as nothing is mentioned about these results in the results section, and I do not find it back in the supplements either (besides describing the results of the systematic search). Did the authors use the H2RA for this cohort instead of background population/non-users; and just pool the results together with the other studies? Didn't this massively affect the results/interpretation?

- Abbreviation RD is not explained. In the legend you mention this is a measure for absolute effects but this is insufficient, and the abbreviation itself should be introduced in the text

- Fig 5 & Supplementary Fig 9 are hard to interpret. Maybe good to add a bit more information in the legend or title. All abbreviations should be explained in legend of each Table/Figure

- So if I interpret these figures correctly, the majority of the cases does occur in the "high risk" groups, yet almost all cases among the "low risk" groups are attributed to PPI use? Meaning, if you stop PPI use, the "low risk" cases would almost all be avoided (assuming causality)? Maybe it is easier to interpret if you also present attributable fractions and/or attributable risks etc? I think you have everything to calculate these – and maybe this is already what is presented?

- Supplementary Table 37: how is "longstanding illness" defined? Is this none/any or is this numerical? All using PPIs will have a longstanding illness – so if it is none/any this may reflect indication of use, not actual use

- Title/legend of Suppl Fig 2 could also be more informative to help the reader interpret the findings. There is no legend explaining the difference between the blue and red lines, and abbreviation PPI is not explained

- Please do use terminology microbiome/microbiota, not microflora- "flora" is for plants not microbes (although used incorrectly in the past)

Minor:

- it seems more logic to first put all tables and then all figures in the supplement instead?

- Better to use "sex" instead of gender as you describe biological differences and not differences related to gender identity

Reviewer #2 (Remarks to the Author):

TITLE

OK.

ABSTRACT

Please confirm in the Instructions for Authors section of the Journal whether the abstract needs to be structured (that is, including the standard sections: Introduction/Aims, Methods, Results and Conclusions).

82.3%: decimals could be rounded (82%).

A comment (of prudence) is missing noting that statistical association does not necessarily imply causation (due, among other things, to the frequent biases of observational studies). In fact, the authors correctly comment in the Discussion section that the identified associations may be partially or completely due to confounding effects.

KEYWORDS: Consider adding other terms (including the most frequently associated diseases that have been found in the present study).

INTRODUCTION

OK.

METHODS

Please explain better how the multivariate analysis was performed, taking into account the potential confounders for the association between PPI intake and the presence of the studied diseases (mainly for propensity score matching, please provide more detailed information).

RESULTS

Overall, the participants had a mean age between 48.4 and 71.4 at baseline. Please explain the potential reasons for such a big difference among the different cohorts included in the study.

DISCUSSION

The authors correctly point out that it remains unclear whether these associations are causal. In this respect, I suggest emphasizing that, for example, compared with non-PPI users, regular PPI users were more likely to be older, obese, smoking, less physically active, with higher rates of comorbidities and medication usage.

It is essential that the authors clearly emphasize in the Discussion section that a statistically significant difference does not necessarily imply that this difference is clinically relevant. In this sense, the HRs, although highly statistically significant, was of only 1.12-1.54, which translates into a rather small absolute increase in risk.

The authors point out that the performance of the model was moderate to low; in fact, the model could effectively stratify the PPI-related adverse events, but with a RD of only 2.94% for the individuals at the upper 20%. Again, this considerably small association effect should be underlined in the Discussion

section.

The authors state: “We noted that the aforementioned meta-analysis did not include 4 recent cohort or case control studies,33-36 which all demonstrated a significant association between PPI and diabetic risk. A preliminary meta-analysis of all published results suggested similar results as ours”. Please add here the corresponding reference or specific data.

The authors concluded that their risk stratification model provides a feasible practical solution; and that the importance of risk stratification is not only to identify those who are at high risk and take prevention individually, but also to screen patients who could safely use PPIs which in turn, reduce fears and increase treatment adherence among patients. However, it should be noted that it is precisely the patients with the most risk factors (typically older and with comorbidities) who most frequently require treatment with PPIs, and therefore the usefulness of this stratification model is limited.

REFERENCES

OK.

TABLES

OK.

FIGURES

OK.

Reviewer #3 (Remarks to the Author):

The manuscript presents a comprehensive analysis of PPI use and numerous disease outcomes using five different population based cohorts. The study has been conducted well and is reported well. The statistical methods are appropriate and the discussion presents the limitations adequately. My only concern is the overwhelming amount of information presented in one paper. It is more difficult to read because the authors need to explain details for each cohort, along with the complex analyses undertaken. The supplementary material is exhaustive. I can see the value in presenting all this comprehensively however it feels far too much for one paper. Perhaps this is something for the journal and authors to consider and weigh up. I can see why the authors would not want to lose any of the information presented but it becomes difficult to see how all the results fit together across many analyses and many outcomes.

Reviewer #4 (Remarks to the Author):

Review of “Personalized Use of Proton Pump Inhibitors: Safety Profile and Risk Stratification for High-Burden Diseases in a Combined Analysis of Five Population-Based Cohorts”

General comments

This is an interesting manuscript that aims at evaluating the relationship between self-reported exposure to Proton Pump Inhibitors and risk of major chronic diseases. The study is a pooling analysis of 5 large population studies. The results are novel and have a promising potential to provide new evidence on a candidate risk factor for several chronic conditions, and to inform on public health decisions. Despite several positive elements, the current version of the manuscript has three limitations. First, the current version of the text contains several instances where describe scientific evidence is not described appropriately. Some of the sentence either read vague or use inappropriate wording. Please refer to the detailed comments, two examples are the way concepts like ‘risk stratification’ and ‘effects’ are used in the text. Second, the information on the prevalence of chronic conditions in Figure 1 should include the frequency of chronic conditions in each cohort, rather than reporting the number of participants for specific events. Third, little information is provided on the overall and cohort-specific prevalence of PPI exposure in this study. Also, there is no mention of potential exposure misclassification on a binary covariate on the observed results in the discussion.

Detailed comments

- Title: why personalized, can the Authors justify the use of this word?
- Line 60: Please define the expression “PPI-related absolute risks ...” – What do the Authors mean? What risk are they referring to?
- Line 79: Can Authors justify the use of the word “appropriate”?
- Line 80: Please consider revision of the expression “Individualized reducing ...”. The expression is not informative.
- Line 87: Consider providing some context to the sentence “Risk stratification has been reported ... “. The sentence currently lacks clarity.
- Line 95: Consider spelling cohort names out. Though I suspect this is due to the fact that Methods are at the end of the manuscript?
- Line 137: replace revealed with indicated.
- Line 138: Please avoid using expressions like ‘evident’ dose -response. What does evident indicate?
- Line 139: this is an observational setting that can at best estimate associations. The word effects suggest a causal relationship which is beyond the ambition on the current study. I suggest to replace the word effects throughout the manuscript with associations or relationships.
- Line 150: not really clear what a more comprehensive confounder control indicate. Please use less vague expressions.
- Line 151: the sentence in item 3 does not read meaningful.
- Line 154: avoid using the word effects.
- Line 155: replace diabetic risk with risk of developing type-2 diabetes.
- Line 156: a reference to the meta-analysis is missing.
- Lines 157 to167: the text is potentially relevant but it is very dense but lacks specificity: it addresses multiple potential morbid conditions and candidate mechanisms at once.
- Line 170: what is indication for using PPIs?
- Line 171: add ... confounders .. in statistical models.

- Line 173: replace increased risk with positive associations.
- Line 181: The sentence reading “our risk stratification model provides ...” is unclear. I wonder whether the Authors have a correct interpretation of what risk stratification represents. It is not a statistical model. Please consider revising the current version of the text.
- Line 186: what is safety profile of PPIs?
- Line 191: Please consider revising the text, science is not about being confident about results.
- Line 195: It is very informative that exposure consistency was evaluated across cohorts. What about exposure accuracy? In other words, was PPI exposure subject to misclassification. And if so, can one anticipate the level and the direction of bias?
- Line 196: the use of random effects is not a limitation.
- Line 202 to 212: this is a useful paragraph. However, it is not obvious to understand how the results of the current study informed on the conclusions in this paragraph.
- Line 216: the sentence on risk stratification reads unclear.
- Line 319: why were these participants excluded? It would be informative to model them.
- Line 360: Please specify what was the primary time variable in Cox models.
- Line 361-362. The strategy used does not comply with a formal definition of confounders. Confounders also need to show some association with the main exposure(s).
- Line 375: the lagged analyses does not read clear enough.
- In tables and figures, please make more informative use of footnotes to clarify important features of the quantities reported.
- Table 1: postmenopausal (%), specify ... among women?
- T1: add alcohol to “Never drinkers”
- T1: Comorbidities: Are these prevalent conditions? Please clarify.
- T1: Please add information about PPI frequency and duration.
- Figure 1, the flowchart: it is not entirely readable.
- F1: replace baseline for basic (population).
- F1, the blue box: the reported N should indicate the number of incident chronic conditions, rather than the participants for specific events.
- F1: frequencies reported here are not consistent with T1.

Reviewer Comments:

Reviewer #1:

1. I read with interest the paper examining adverse events in PPI users using several large previously published cohorts. The authors included a novel approach to assess the multitude of outcomes associated with PPI use, and these results are surprisingly consistent among the different study cohorts, with dose-response curves supporting the proposed causal association for almost all outcomes. This article also includes the construction of a prediction model, to predict who is most at risk of any of these 15 outcomes (and for whom discontinuation should especially be considered?), yet I do wonder if there shouldn't be any efforts to discontinue treatment in all receiving inappropriate PPI use.

Response: Thank you very much for your positive feedback on this paper.

The question of discontinuing PPI treatment for all individuals with inappropriate use is indeed complex, as it necessitates a delicate balance between the benefits and risks of these medications. While our study highlights the potential adverse effects associated with PPIs, it's crucial to recognize the therapeutic value of PPIs in managing specific acid-related disorders.

In this context, we introduced a risk stratification approach for personalized use of PPIs. Its primary purpose is to identify individuals at a higher net risk of experiencing adverse events linked to PPI usage. This risk stratification allows for more precise interventions. For individuals at higher net risk, effective strategies may include dose reduction, discontinuation, switching to 'on-demand' use, adopting less potent acid suppressants like H2RAs, and regular monitoring for early signs of adverse events.

Besides, clinical judgment should remain a critical factor in decision-making to striking a balance between delivering effective treatment for acid-related disorders and minimizing the potential risks of prolonged PPI use. Further research are still needed to refine the criteria for discontinuing inappropriate PPI treatment based on an individual's unique risk profile.

2. The paper is interesting, yet there is a lot of information in there, with the estimation of effects prediction model and also some sort of evidence synthesis (which is not described in detail yet seems to have caught many/most of the relevant and recent papers). The authors also succeeded in properly describing the problem and potential the underlying mechanisms – and the methodology seems appropriate

Response: Thank you. In our revised manuscript, we have expanded the Methods section to provide a more detailed description of the prediction model construction, data analysis. We hope this additional information will further enhance the clarity and comprehensiveness of our study.

3. I think the part on the prediction model and how to interpret/use it, is a bit hard to follow (because of limited word count and many important parts moved to the supplements).

Response: Thank you. We added the process of prediction model development in the Methods section. In the study, we aimed to develop a clinical prediction model to predict the risk of a composite outcome of PPI-related high-burden diseases. The candidate predictors were considered

as established factors influencing the incidence of any of the 15 outcomes and are typically obtained through questionnaires, such as age, body mass index, and smoking status. We fitted Cox proportional hazards models to estimate the coefficients associated with each potential risk factor for the first diagnosis of the composite outcome (i.e., any of the 15 PPI-related diseases). Variables were retained in the prediction model if they exhibited a hazard ratio of <0.85 or >1.15 (for binary variables) and achieved statistical significance at the 0.01 level. The prediction model was initially derived from participants in England and subsequently geographically validated in participants from Scotland and Wales. Discrimination of the prediction model was evaluated using C-index over 1, 5, and 10 years, while calibration was assessed using calibration plots based on risk deciles at 5 years. Additionally, we constructed a nomogram for the model to offer a more direct means of assessing the 1-, 5-, and 10-year probability of any of the 15 unintended outcomes. In the Results section, we reported these variables included in the final models and showed the performance and nomogram of the final models in the Supplementary file.

4. Regarding clinical implications: Those who are not at high risk now, may be in a few years - and it seems almost all cases of these adverse events in the low risk groups are attributed to PPI use. In absolute numbers, these outcomes are indeed most frequent in the "high risk" but relatively speaking, the risks do not seem negligible in the low-risk groups? The authors did not specifically mention the inappropriate over-use of PPIs which is estimated to be up to 70% in a Cochrane review.

Response: Thank you very much. We agree with you that individuals categorized as low risk presently may transition into higher-risk categories over time. So we suggest the evaluation of future PPI-related risk should be carried out regularly.

Regarding the cases of these adverse events in the low risk groups, maybe you are misled by the previous figure 5 which are not very clear. The "443/497 (89.1%)" are the annual number of cases attributed to PPI use in the subpopulation (baseline risk 50-100%)/total PPI related cases. Not all cases of these adverse events in the low risk groups are attributed to PPI use. We revised the figure 5 to avoid ambiguity.

Regarding the risks in the low-risk groups, we agree that they do not seem negligible. Approximately 10.9% of cases happened in the lower 50% population by baseline risk. In clinical practice, the decision should be based on the risk and benefits for individual patients. We highlighted this point in the implications.

For the rate of inappropriate overuse of PPIs, we cited this Cochrane review ([doi: 10.1002/14651858.CD011969.pub2](https://doi.org/10.1002/14651858.CD011969.pub2)) and emphasized the significance of addressing this concern. You may see the corresponding revision below:

Implication for clinical practice and research

"Given the links with a substantial global disease burden and the high rate of inappropriate overuse of PPIs (up to 70%),⁵¹ the potential impact of long-term PPI use should not be ignored, even if the causal effects for these outcomes have not been established. In practice, the net risk of PPI-related adverse effects is low in those with low baseline risk, but it is not negligible and the

risk profile may change over time. Personalized prevention is feasible by regular evaluating the baseline risk with readily available predictive factors for long-term PPI users while focusing on the high-risk patients. For high-risk individuals, potential effective strategies, such as dose reduction, discontinuation, transitioning to “on-demand” use, considering less profound acid suppressants like H2RAs, and regular monitoring for early indications of adverse events (e.g., blood glucose levels for the risk of diabetes ^{9,10}), may help mitigate the additional absolute risk associated with PPI use.”

5. I do hope this paper helps in the awareness among clinicians, as a broad range of outcomes should be considered when prescribing PPIs, ideally restricted to well-established indications. There has been a lot of discussion on the harm vs. benefit of PPIs, and the risks of individual outcomes – but the present paper does a good attempt to make a composite outcome – although this still seems likely/plausible to be an underestimation of the true effect (assuming causality) as some analysis seem underpowered?

Response: Thank you very much.

We agree with you that a composite outcome would be helpful to evaluate the overall harms and benefits, however some analysis might be underpowered. We acknowledge in study limitations that some analyses in the present study might be underpowered. We also emphasized the need for additional research with larger sample sizes to provide more robust insights into the potential effects of PPIs on these outcomes.

"Fifth, despite our study's large sample size and extensive range of outcomes, it may still be underpowered for certain associations, such as the one between PPI use and gastric cancer. Additionally, as not all outcomes were available in every cohort, analyses for 'Low back pain' and 'Blindness and vision loss' still rely on data from a single cohort, which limits our ability to assess dose-response relationships for PPIs and conduct comparisons with a positive control. Further research with larger sample sizes is needed to strengthen the evidence regarding the potential effects of PPIs on these specific outcomes."

6. I think the article could use a discussion on the association with gastric cancer. Many studies have been published on this association, even a handful of meta-analyses (some mentioned in supplements). I was first not sure if it was not in that list of 30-35 diseases, or that it didn't come out as significant. Digging into the data, I found that it was the latter scenario – and it is remarkable that the pooled effect size of gastric cancer is lower than many other (significant) associations. This may suggest that this large study is still underpowered for some associations – and that it is “even worse” than the presented association with 15 important diseases.

Response: Thank you. We have added a discussion about the association of PPI use with gastric cancer in the revised manuscript. You can find it below:

"Furthermore, our study failed to find the significant association between PPI use and gastric cancer as reported by the two recent meta-analysis.^{37,38} While the direction of the pooled association in our study aligns with these prior reports, the lack of statistical significance may suggest that this large-scale study might still be underpowered for certain associations. This also

raises the possibility that the situation may be even more challenging than the associations presented for the 15 important diseases."

We have also acknowledged this limitation in the revised manuscript:

"Fifth, despite our study's large sample size and extensive range of outcomes, it may still be underpowered for certain associations, such as the one between PPI use and gastric cancer..."

7. I would put Supplementary Table 37 and Suppl Fig 7 as main Table.

Response: Thank you. We appreciate your suggestion to integrate Supplementary Table 37 and Supplementary Figure 7 into the main body of the manuscript.

Supplementary Figure 7 visually represents the application of the predictive model derived from Supplementary Table 37 for risk stratification. It's important to note that the primary objective of introducing the risk stratification method in this study is to identify high-risk individuals for more targeted PPI use recommendations, rather than solely emphasizing the construction of the model itself. This aligns with the broader scope of our study, which extensively investigates potential health risks associated with regular PPI use across diverse population cohorts.

After careful consideration of the manuscript's overall coherence, we have opted to incorporate Supplementary Table 37 as the main table within the manuscript. Supplementary Figure 7, being an integral part of the risk stratification method, remains in the Supplementary file. Additionally, we have enhanced the details regarding the model parameters in the results section to provide a more comprehensive understanding.

8. The prediction model is of great clinical importance as it could guide in whom discontinuation of PPI treatment should be most beneficial. I would at least mention which variables ended up in the prediction model (in the results section).

Response: Thank you for this suggestion. In the revised manuscript, we have included a description of the variables included in the prediction model within the results section.

"For easier screening of high-risk individuals during clinical practice, we developed a prediction model that incorporates several key variables as predictors for a composite outcome of any of the 15 diseases. These predictors include age, BMI, the number of treatments or medications taken, smoking status, longstanding illness, overall health rating, and self-reported usual walking pace. Detailed information about these variables in the prediction model were presented in Table 2..."

9. I am not sure how the authors applied the H2RA active comparator design, as nothing is mentioned about these results in the results section, and I do not find it back in the supplements either (besides describing the results of the systematic search). Did the authors use the H2RA for this cohort instead of background population/non-users; and just pool the results together with the other studies? Didn't this massively affect the results/interpretation?

Response: Thank you. We apologize for not providing detailed information in the original manuscript. We performed an active comparator design using H2RAs (a class of less profound

acid suppressants) as the control group within the CDARS. We did not use such analysis in other cohorts because the number of H2RA users are too small.

The rationale behind using an active comparator (H2RAs) in this context was to provide more robust evidence by comparing PPI users to a group that closely resembles them in terms of clinical indication, rather than relying solely on non-users from the background population. This helps mitigate potential bias related to the clinical indications for which PPIs are prescribed, and pooling the results from active comparator design with other estimates to get the overall estimates could lead to more conservative results. We also noted that the results obtained through the H2RA active comparator design align closely with the associations found in other cohorts for almost all outcomes associated with PPI use. Furthermore, we performed a sensitive analysis by pooling the estimates of other four cohorts after excluding the CDARS population, which showed consistent results as the primary results.

We have expanded the methods section in the revised manuscript to provide a detailed description of how the H2RA active comparator design was applied in the CDARS database. Additionally, we have included descriptions of the main results obtained through the active control design in the revised results section.

Methods:

“For the UK biobank, NHS, NHS II, and HPFS, we considered the non-regular PPI users as control. For CDARS, we used an active comparator design, taking H2 receptor antagonists (H2RAs), a class of less profound acid suppressants, as the control. This helps mitigate potential bias related to the clinical indications for which PPIs are prescribed. We performed an outcome-wide approach to identify diseases associated with PPI use in each cohort, and then pooled the associations with meta-analyses to get the overall estimates, which may lead to more conservative results.”

Results:

“...The results obtained through the H2RA active comparator design in the CDARS database align closely with the associations found in other cohorts for almost all outcomes associated with PPI use...”

“The primary results did not reveal major changes in the sensitivity analyses by lagging the exposure for 4 years, using propensity score analysis, and excluding CDARS”

10. Abbreviation RD is not explained. In the legend you mention this is a measure for absolute effects but this is insufficient, and the abbreviation itself should be introduced in the text.

Response: Thank you. We have introduced and explained the abbreviation RD, which stands for Risk Difference, in the main text for clarity. We have also revised the legend to provide a more comprehensive explanation of this measure as follows:

“We evaluated the HRs of PPI use as compared with no-PPI use in each quartile group, then translated to absolute effects (risk differences, RDs) associated with PPI use at one year by the method described by Altman.⁵¹”

Reference:

VanderWeele, T.J. & Ding, P. *Sensitivity Analysis in Observational Research: Introducing the E-Value*. *Ann Intern Med* 167, 268-274 (2017).

11. Fig 5 & Supplementary Fig 9 are hard to interpret. Maybe good to add a bit more information in the legend or title. All abbreviations should be explained in legend of each Table/Figure

Response: Thank you. We have revised Fig 5 and Supplementary Fig 9 to enhance clarity. The legends for both figures now provide additional information to facilitate interpretation. We have also explained all abbreviations in the legend for better comprehension. Please find the revised figures and legends below:

“**Figure 5.** Absolute risk of PPI-related high-burden diseases according to the distribution of the baseline predicted risk.

Abbreviation: AF, attributable fraction; PPI, proton pump inhibitor; RD, risk difference.

This histogram presented the distribution of baseline predicted risk for any of the 15 PPI-related diseases. The performance and nomogram for the prediction model is available in Supplementary Table S38 and Supplementary Fig. S7-8. The RD and AF of PPI use for one year was calculated. Annual number of cases attributed to PPI use in each strata (i.e., n) was calculated based on attributable risk and exposure, and then summed as the total annual number of cases attributed to PPI use in all populations (i.e., total). The results showed that most cases were occurred in the individuals with high baseline predicted risk, and those who with low baseline risk do not need to be over panicked and should adhere to PPI treatment.”

“Supplementary Figure 9. Sensitivity analysis: PPI-related absolute risk for major high-burden diseases according to the distribution of the baseline predicted risk after excluding osteoarthritis.

Abbreviation: AF, attributable fraction; PPI, proton pump inhibitor; RD, risk difference. This histogram presented the distribution of baseline predicted risk for any of the 15 PPI-related diseases. The performance and nomogram for the prediction model is available in Supplementary Table S38 and Supplementary Fig. S7-8. The RD and AF of PPI use for one year was calculated. Annual number of cases attributed to PPI use in each strata (i.e., *n*) was calculated based on attributable risk and exposure, and then summed as the total annual number of cases attributed to PPI use in all populations (i.e., total). The results showed that most cases were occurred in the individuals with high baseline predicted risk, and those who with low baseline risk do not need to be over panicked and should adhere to PPI treatment.”

12. So if I interpret these figures correctly, the majority of the cases does occur in the “high risk” groups, yet almost all cases among the “low risk” groups are attributed to PPI use? Meaning, if you stop PPI use, the “low risk” cases would almost all be avoided (assuming causality)? Maybe it is easier to interpret if you also present attributable fractions and/or attributable risks etc? I think you have everything to calculate these – and maybe this is already what is presented?

Response: Thanks for your suggestion. We have calculated the attributable fractions of PPI use over one year and added in the Fig 5 & Supplementary Fig 9. In fact, many factors, like smoking and obesity, contributed to the risk of high-burden diseases, and PPI only accounted for a relative small proportion (AF<5%), so the cases would not all be avoided after stopping PPI use in low risk group. We reported the annual number of cases attributed to PPI use in strata, which was small proportion of all cases occurred in each strata, and those case would almost all be avoided if stopping PPI use. We revised the legend of Fig 5 & Supplementary Fig 9 to remove confusion.

13. Supplementary Table 37: how is “longstanding illness” defined? Is this none/any or is this numerical? All using PPIs will have a longstanding illness – so if it is none/any this may reflect indication of use, not actual use.

Response: Thank you. In the UK Biobank, the “longstanding illness” variable is defined as a binary response to the question, “Do you have any long-standing illness, disability, or infirmity?”

It's recorded as a binary variable (No/Yes), indicating the presence (any) or absence (none) of a longstanding illness.

In the UK Biobank cohort, PPI users indeed have a higher proportion of individuals with a longstanding illness compared to non-users. This could reflect that individuals prescribed PPIs might have an underlying medical condition leading to their prescription, which may reflect the indication for use rather than the actual use of PPIs. But in Supplementary Table 37, this variable itself is simply included as one of the predictive factors in our model and does not imply a direct relationship with PPI use.

14. Title/legend of Suppl Fig 2 could also be more informative to help the reader interpret the findings. There is no legend explaining the difference between the blue and red lines, and abbreviation PPI is not explained.

Response: Thank you for this suggestion. In the revised manuscript, we have updated the title of Supplementary Figure 2 to be more informative and to help readers interpret the findings. We also added a brief legend and explained the abbreviation PPI. You may see below:

“Supplementary Figure 2. Value of the joint minimum strength of association on the risk ratio scale that an unmeasured confounder must have with the exposure and the outcome to fully explain away an observed HR between PPI and the outcome.

Abbreviation: PPI, proton pump inhibitor. The blue line represents e-values, and the red line represents the lower limit of e-values.”

15. Please do use terminology microbiome/microbiota, not microflora- “flora” is for plants not microbes (although used incorrectly in the past)

Response: Thank you. In the revised manuscript, we have replaced the term “microflora” with “microbiota” to accurately reflect the microbial community associated with the human body.

Minor

16. it seems more logic to first put all tables and then all figures in the supplement instead?

Response: Thank you for this suggestion. We have reorganized the supplementary materials to present all tables first, followed by the figures.

17. Better to use “sex” instead of gender as you describe biological differences and not differences related to gender identity.

Response: Thank you. We have revised the terminology in the manuscript to use “sex” instead of “gender” when describing biological differences.

Reviewer #2:

1. TITLE: OK.

Response: Thank you.

2. ABSTRACT: Please confirm in the Instructions for Authors section of the Journal whether the abstract needs to be structured (that is, including the standard sections: Introduction/Aims, Methods, Results and Conclusions).

Response: Thank you. We have reviewed the Instructions for Authors section of the Journal. The requirements for the abstract in this journal involve submitting an abstract of approximately 150 words to provide a general introduction to the topic and a brief non-technical summary of the main results and their implications. There is no specific requirement for a structured abstract. Our current abstract complies with the journal's instructions for the final submission.

3. ABSTRACT: 82.3%: decimals could be rounded (82%).

Response: Thank you. We have rounded the percentage to 82% in the final version of the abstract.

4. ABSTRACT: A comment (of prudence) is missing noting that statistical association does not necessarily imply causation (due, among other things, to the frequent biases of observational studies). In fact, the authors correctly comment in the Discussion section that the identified associations may be partially or completely due to confounding effects.

Response: Thank you. We agree with you that statistical association does not necessarily imply causation, especially in the context of observational studies. We have added a comment to the abstract to address this concern as follows:

“While statistical association does not necessarily imply causation, its potential safety concerns suggests that personalized use of PPIs through risk stratification might guide appropriate decision-making for patients, clinicians, and the public.”

5. KEYWORDS: Consider adding other terms (including the most frequently associated diseases that have been found in the present study).

Response: Thank you. We have added the terms “diabetes”, “respiratory infections”, and “chronic kidney disease” to the keywords. These diseases are not only significantly associated with PPI use in our study but are also frequently reported to be closely related to PPI use. Additionally, they represent diseases with a substantial burden.

6. INTRODUCTION: OK.

Response: Thank you.

7. METHODS: Please explain better how the multivariate analysis was performed, taking into account the potential confounders for the association between PPI intake and the presence of the studied diseases (mainly for propensity score matching, please provide more detailed information).

Response: Thank you. We have provided more detailed information on how the multivariate analysis was performed in the revised methods section. Here is the updated content:

“For the three US cohorts, we applied multivariable time-dependent Cox regression models stratified by age and time period (in 2-year intervals) and additionally adjusted for ethnicity, BMI, smoking status, alcohol consumption, physical activity, overall diet quality (AHEI-2010), portions of fruit and vegetable intake, family history of specific diseases, clinical indication for PPI use (i.e., GERD, gastric or duodenal ulcer, gastrointestinal bleeding), medications (multivitamin use, NSAID, aspirin, statin, ACEIs, beta-blockers, calcium-channel blockers, thiazide diuretics, metformin, antibiotic, oral steroids), comorbidities (diabetes, hypertension, hypercholesterolaemia), and female-specific indicators (i.e., parity, menopausal status, and postmenopausal hormone use, for NHS and NHS II only). For the UK Biobank, we stratified the analyses jointly by age, sex, and UK assessment centres, adjusting for similar variables (i.e., demographic factors, lifestyle habits, medications, comorbidities, PPI clinical indications) as in the US cohorts, and additionally adjusted for self-reported overall health rating and longstanding illness. In the multivariable Cox regression models for the CDARS database, we stratified by age and sex, and additionally adjusted for the medications, comorbidities, and PPI indications mentioned above.”

We also detailed the propensity score method as follows:

“...Second, we carried out a propensity score (PS) analysis using the inverse probability treatment weighting (IPTW) method in the UK Biobank and CDARS database to adjust for potential bias in the allocation of patients to PPI. The propensity score was estimated using a logistic regression model which included all of the aforementioned covariates as potential predictors for PPIs. A weight was then calculated for each patient as $1/PS$ in the high adherence group and $1/1-PS$ for those in the non-PPI user groups (UK Biobank) or H2RA user groups (CDARS database). Extreme weight values were truncated at the 5th and 95th percentile ends of the distribution. We confirmed that the IPTW method (through weighting) had adequately balanced the covariate profile of the two groups by comparison of the unweighted and weighted standardized difference in means/proportions for each covariate.”

8. RESULTS: Overall, the participants had a mean age between 48.4 and 71.4 at baseline. Please explain the potential reasons for such a big difference among the different cohorts included in the study.

Response: Thank you. The variations in the mean age among the different cohorts included in our study might be attributed to the differences in the population characteristics of each cohort. Each cohort represents a distinct population with its unique demographics and recruitment strategies.

For example, the NHS (age 30–55 in 1976) and NHS II (age 25–42 in 1989) cohorts mainly consist of female registered nurses. The HPFS began in 1986 with the enrollment of male health professionals aged 40–75 years. The UK Biobank includes a broader sample of the general UK population aged 37–73 at baseline (recruited from 2006 to 2010). The CDARS database encompasses a diverse Hong Kong population with various age groups, contributing to the diversity of the study population. The age differences reflect the diverse nature of the included cohorts. When performing analyses, we accounted for age differences by stratification or adjustment, ensuring that our results are appropriately adjusted for age-related factors.

9. DISCUSSION: The authors correctly point out that it remains unclear whether these associations are causal. In this respect, I suggest emphasizing that, for example, compared with non-PPI users, regular PPI users were more likely to be older, obese, smoking, less physically active, with higher rates of comorbidities and medication usage.

Response: Thank you. we have emphasized this limitation by adding the following statement to the Discussion section:

“First, owing to the nature of observational study, we could not ratify the causal relationship. This is particularly noteworthy as regular PPI users, in comparison to non-users, were more likely to be older, obese, smokers, less physically active, and had higher rates of comorbidities and medication usage. Although we made efforts to control for various confounders, residual confounding remains a possibility.”

10. DISCUSSION: It is essential that the authors clearly emphasize in the Discussion section that a statistically significant difference does not necessarily imply that this difference is clinically relevant. In this sense, the HRs, although highly statistically significant, was of only 1.12-1.54, which translates into a rather small absolute increase in risk.

Response: Thank you. We agree with you that statistical significance does not always imply clinical relevance, especially considering the relatively modest HRs ranging from 1.12 to 1.54 in our study. We have added this consideration into the revised Discussion section with the following statement:

“However, statistical significance does not always indicate clinical relevance, especially with the relatively modest HRs ranging from 1.12 to 1.54 in our study. The PPI-associated absolute increase in risk, while statistically significant, should be interpreted cautiously in terms of clinical significance.”

11. DISCUSSION: The authors point out that the performance of the model was moderate to low; in fact, the model could effectively stratify the PPI-related adverse events, but with a RD of only 2.94% for the individuals at the upper 20%. Again, this considerably small association effect should be underlined in the Discussion section.

Response: Thank you. In the revised discussion section, we have emphasized the modest effect of PPI-related adverse events by adding the following statement:

“The PPI-associated absolute increase in risk, while statistically significant, should be interpreted cautiously in terms of clinical significance. In our prediction model that effectively stratifies the PPI-related adverse events, the absolute risk difference (RD) was 2.94% for individuals at the upper 20% of the baseline predicted risk, suggesting the degree of effect was small to modest in clinical practice.”

12. DISCUSSION: The authors state: “We noted that the aforementioned meta-analysis did not include 4 recent cohort or case control studies,33-36 which all demonstrated a significant

association between PPI and diabetic risk. A preliminary meta-analysis of all published results suggested similar results as ours". Please add here the corresponding reference or specific data.

Response: Thank you. We have updated the meta-analysis by incorporating the results from four recent cohort or case-control studies (references 33-36), which were not included in the aforementioned meta-analysis. The pooled results, illustrated in Figure 1 below, combine data from these new studies with the previously published ones, demonstrating a significant association between PPI use and the risk of diabetes. The meta-analysis yielded a summary hazard ratio (HR) of 1.17 (95% CI: 1.01-1.34), indicating a 17% increase in the risk of diabetes among PPI users compared to non-users, with high heterogeneity ($I^2=97\%$).

We have added the corresponding data to the statement in the revised discussion section, as follows:

"...A preliminary meta-analysis of all published results (HR=1.17, 95%CI:1.01-1.34, heterogeneity: $I^2=97\%$) suggested similar results as ours."

Figure 1. Forest plot of PPI use and risk of developing type-2 diabetes based on aforementioned meta-analysis and four new studies using random-effects model.

References:

- Barnett M, Argo T, Alexander B, Perry P. A regional comparison of developing diabetes among VA patients exposed to typical and atypical antipsychotics relative to corticosteroids and proton pump inhibitors. *Annals of clinical psychiatry : official journal of the American Academy of Clinical Psychiatrists.* 2006;18(1):1-7.
- Blackburn D, Hux J, Mamdani M. Quantification of the Risk of Corticosteroid-induced Diabetes Mellitus Among the Elderly. *Journal of general internal medicine.* 2002;17(9):717-20.
- He Q, Yang M, Qin X, Fan D, Yuan J, Pan Y. Risk stratification for proton pump inhibitor-associated type 2 diabetes: a population-based cohort study. *Gut.* 2021;70(11):2212-3.
- Lin HC, Hsiao YT, Lin HL, Uang YS, Cheng HW, Wang Y, et al. The use of proton pump inhibitors decreases the risk of diabetes mellitus in patients with upper gastrointestinal disease: A population-based retrospective cohort study. *Medicine (Baltimore).* 2016;95(28):e4195.

Moayyedi P, Eikelboom JW, Bosch J, Connolly SJ, Dyal L, Shestakovska O, et al. Safety of Proton Pump Inhibitors Based on a Large, Multi-Year, Randomized Trial of Patients Receiving Rivaroxaban or Aspirin. *Gastroenterology*. 2019;157(3):682-91.e2.

Yuan J, He Q, Nguyen LH, Wong MCS, Huang J, Yu Y, et al. Regular use of proton pump inhibitors and risk of type 2 diabetes: results from three prospective cohort studies. *Gut*. 2021;70(6):1070-7.

Kuo HY, Liang CS, Tsai SJ, Chen TJ, Chu CS, Chen MH. Dose-Dependent Proton Pump Inhibitor Exposure and Risk of Type 2 Diabetes: A Nationwide Nested Case-Control Study. *Int J Environ Res Public Health*. 2022;19(14).

Loosen SH, Kostev K, Luedde M, Qvartskhava N, Luedde T, Roderburg C. Long-term use of proton pump inhibitors (PPIs) is associated with an increased risk of type 2 diabetes. *Gut*. 2022;71(8):1687-8.

Ciardullo S, Rea F, Savaré L, Morabito G, Perseghin G, Corrao G. Prolonged Use of Proton Pump Inhibitors and Risk of Type 2 Diabetes: Results From a Large Population-Based Nested Case-Control Study. *J Clin Endocrinol Metab*. 2022;107(7):e2671-e9.

Czarniak P, Ahmadizar F, Hughes J, Parsons R, Kavousi M, Ikram M, et al. Proton pump inhibitors are associated with incident type 2 diabetes mellitus in a prospective population-based cohort study. *Br J Clin Pharmacol*. 2022;88(6):2718-26.

13. DISCUSSION: The authors concluded that their risk stratification model provides a feasible practical solution; and that the importance of risk stratification is not only to identify those who are at high risk and take prevention individually, but also to screen patients who could safely use PPIs which in turn, reduce fears and increase treatment adherence among patients. However, it should be noted that it is precisely the patients with the most risk factors (typically older and with comorbidities) who most frequently require treatment with PPIs, and therefore the usefulness of this stratification model is limited.

Response: Thank you very much. We agree with you that the patients with the most risk factors are likely to be those who most frequently require treatment with PPIs.

Our risk stratification method may provide a way to evaluate patients' future PPI-related risk and guide personalized decision-making. Stopping PPI is not the only action the high-risk patients could consider. In addition, these patients may also consider dose reduction, transitioning to “on-demand” use, considering less profound acid suppressants like H2RAs, and regular monitoring for early indications of adverse events (e.g., blood glucose levels for the risk of diabetes). These actions can still be considered by those require PPI treatment. This model is not intended to discourage PPI use in high baseline risk group who genuinely need it but rather to inform and guide personalized decision-making in clinical practice. This risk stratification approach has been employed in screening strategies for other medications, such as statins, where individuals at higher risk for diabetes due to statin use are identified and monitored closely.¹ We explained this clearer in the discussion.

“Our risk stratification approach provides a feasible practical solution. Risk stratification, in this context, facilitates identifying individuals at higher or lower baseline risk for PPI-associated adverse events. This involves a comprehensive assessment of individual characteristics and health

status through a prediction model. The importance of risk stratification is not only to identify those who are at high risk and take preventive measures individually to minimize the additional harm caused by long-term PPI use, but also to screen patients who could safely use PPIs. This, in turn, reduces fears and increases treatment adherence among patients.”

Reference:

1. Mansi IA, Sumithran P, Kinaan M. Risk of diabetes with statins. *BMJ* 2023;381:e071727

14. REFERENCES: OK.

Response: Thank you.

15. TABLES: OK.

Response: Thank you.

16. FIGURES: OK.

Response: Thank you.

Reviewer #3:

1. The manuscript presents a comprehensive analysis of PPI use and numerous disease outcomes using five different population based cohorts. The study has been conducted well and is reported well. The statistical methods are appropriate and the discussion presents the limitations adequately. My only concern is the overwhelming amount of information presented in one paper. It is more difficult to read because the authors need to explain details for each cohort, along with the complex analyses undertaken. The supplementary material is exhaustive. I can see the value in presenting all this comprehensively however it feels far too much for one paper. Perhaps this is something for the journal and authors to consider and weigh up. I can see why the authors would not want to lose any of the information presented but it becomes difficult to see how all the results fit together across many analyses and many outcomes.

Response: Thank you. We appreciate your positive feedback of the study design, methods, and reporting. We understand your concern about the comprehensive nature of the information presented in the manuscript, including the details for each cohort and the complex analyses, which may affect the overall readability. We have carefully considered your suggestion and have made several revisions in the revised manuscript to strike a better balance between comprehensiveness and readability:

1). We have limited the presentation of detailed results to the main manuscript for the pooled analysis of the five cohorts. Additional detailed results for each cohort have been moved to the supplementary materials. 2). A concise summary of the main findings across cohorts has been provided in the Abstract section to assist readers in grasping overarching patterns without delving into intricate details. 3). The supplementary material has been restructured for improved organization, with clearer labeling, reordered tables and figures, and enhanced navigation to make it more accessible for readers interested in specific details

These changes may partially address your concerns by providing a more balanced and reader-friendly presentation. Any additional guidance or specific recommendations to further enhance the clarity and readability of this paper is welcome. Thank you again for your valuable feedback.

Reviewer #4:

1. General comments: This is an interesting manuscript that aims at evaluating the relationship between self-reported exposure to Proton Pump Inhibitors and risk of major chronic diseases. The study is a pooling analysis of 5 large population studies. The results are novel and have a promising potential to provide new evidence on a candidate risk factor for several chronic conditions, and to inform on public health decisions. Despite several positive elements, the current version of the manuscript has three limitations. First, the current version of the text contains several instances where describe scientific evidence is not described appropriately. Some of the sentence either read vague or use inappropriate wording. Please refer to the detailed comments, two examples are the way concepts like ‘risk stratification’ and ‘effects’ are used in the text. Second, the information on the prevalence of chronic conditions in Figure 1 should include the frequency of chronic conditions in each cohort, rather than reporting the number of participants for specific events. Third, little information is provided on the overall and cohort-specific prevalence of PPI exposure in this study. Also, there is no mention of potential exposure misclassification on a binary covariate on the observed results in the discussion.

Response: Thank you. We appreciate your insightful comments and suggestions. We have carefully considered each point raised and made the following revisions to enhance the clarity and precision of our study.

We acknowledge the need for improvement in describing scientific evidence appropriately. In the revised manuscript, we have made more clarity and precision in the language used to convey scientific findings, addressing the concerns you raised. Specifically, we replaced the term “effect” with “associations” or “relationships” to accurately convey the observational nature of our study. Furthermore, we clarified the concept of “risk stratification” to ensure a better understanding of its application in our study. The revised text now reads:

“Risk stratification, in this context, facilitates identifying individuals at higher or lower baseline risk for PPI-associated adverse events. This involves a comprehensive assessment of individual characteristics and health status through a prediction model. The importance of risk stratification is not only to identify those who are at high risk and take preventive measures individually to minimize the additional harm caused by long-term PPI use, but also to screen patients who could safely use PPIs. This, in turn, reduces fears and increases treatment adherence among patients.”

Regarding Figure 1, we have made improvements to enhance clarity, incorporating information on the frequency of chronic conditions in each cohort for a more comprehensive overview.

In response to your inquiry about the prevalence of PPI exposure, we have added the proportion of PPI users in each cohort to Table 1.

Additionally, we appreciate your attention to exposure accuracy. We acknowledged the potential limitation of exposure misclassification, particularly in the UK Biobank cohort where PPI use was self-reported only at baseline. This limitation is now explicitly stated in the manuscript's limitations section, underscoring the possibility of underestimating true effects due to

misclassification. In this study, misclassification could underestimate the true effects, as the control group may include individuals who initiated PPI use during the follow-up period. To mitigate this, we applied a random-effect model to combine PPI associations and obtain overall estimates. We reported this limitation as follows:

“Third, the definition of PPI use and endpoints is not always consistent among the included cohorts. Furthermore, PPI use was only evaluated once at baseline in the UK Biobank, introducing a chance of misclassification during follow-up. Misclassification could underestimate the true effects, as the control group may include individuals who initiated PPI use during the follow-up period. To minimize the potential influence, we combined the effects with random-effect model as other studies.^{7,8}”

Detailed comments

2. Title: why personalized, can the Authors justify the use of this word?

Response: Thank you. We used the term “personalized” in the title to convey the concept that our study aims to provide insights into the individualized risk associated with proton pump inhibitor (PPI) use. The personalized aspect comes from our risk stratification model, which helps identify individuals within the PPI user population who may have a higher baseline risk of developing PPI-associated adverse events. This allows for more informed decision-making by both patients and clinicians regarding the ongoing use of PPIs, considering the potential risks and benefits based on individual characteristics.

3. Line 60: Please define the expression “PPI-related absolute risks ...” – What do the Authors mean? What risk are they referring to?

Response: Thank you very much. Absolute risks refer to Risk Difference (RD), which show the actual degree of absolute risk of experiencing the adverse events associated with PPIs.

Compared with relative risk, absolute risks is a more clinically useful way to present an effect and more useful to make clinical decisions.

For example.

	Event/total		Relative effect (RR)	Absolute effect (RD)
	Exposure group	Control group		
Case 1	10/100	5/100	$(10/100) / (5/100) = 2$	$(10/100) - (5/100) = 5/100$
Case 2	50/100	25/100	$(50/100) / (25/100) = 2$	$(50/100) - (25/100) = 25/100$

In case 1 and case 2, the absolute effects reflected the true differences (5/100 vs 25/100), but the relative effect could not show the difference (both are 2)

We directly show the absolute effect term here to avoid ambiguities

“PPI-related absolute risks (risk difference, RD) increased with baseline risks,....”

4. Line 79: Can Authors justify the use of the word “appropriate”?

Response: Thank you. The use of the term "appropriate" in the sentence refers to the judicious and justified use of proton pump inhibitors (PPIs) in clinical practice. It implies the need to use PPIs in situations where their benefits outweigh the risks. For example, reducing unnecessary use of PPIs or considering alternative antacid medications to manage gastrointestinal symptoms may be deemed appropriate based on individual characteristics, as suggested by Barbara et al.¹

Reference:

1. Farrell B, Lass E, Moayyedi P, et al. Reduce unnecessary use of proton pump inhibitors. *BMJ* 2022;379:e069211.

5. Line 80: Please consider revision of the expression "Individualized reducing ...". The expression is not informative.

Response: Thank you for this suggestion. We have revised the sentence as follows:

"The need for personalized strategies for reducing unnecessary PPI use has become an urgent subject to be addressed."

6. Line 87: Consider providing some context to the sentence "Risk stratification has been reported ...". The sentence currently lacks clarity.

Response: Thank you. We have expanded the context surrounding the sentence "Risk stratification has been reported..." to provide a clearer understanding. The revised text now reads:

"Thus, individualized treatment based on patients' underlying risk may confer benefits and reduce harms. Such a risk stratification approach, successfully implemented in selecting patients for antihypertensive and statin therapy,^{11,12} has also been applied to individuate avoidance of additional risks related to PPI use, such as type 2 diabetes,⁷ stroke,¹³ and cholelithiasis.⁸ However, its application for other PPI-associated adverse events remains unclear."

7. Line 95: Consider spelling cohort names out. Though I suspect this is due to the fact that Methods are at the end of the manuscript?

Response: Thank you for this suggestion. We have spelled out the full names of the cohorts when they first appear in the results section and subsequently used the abbreviated form in the methods section for brevity. You may see below:

"A total of 2 079 724 participants from UK Biobank (n=501 109), Nurses' Health Study (NHS, n=91 708), NHS II (n=99 641), Health Professionals Follow-Up Study (HPFS, n=30 933), and Clinical Data Analysis and Reporting System (CDARS, n=1 356 333) were included as the basic population for the current analyses."

8. Line 137: replace revealed with indicated.

Response: Thank you. We have replaced the word "revealed" with "indicated" in the revised manuscript.

9. Line 138: Please avoid using expressions like 'evident' dose -response. What does evident indicate?

Response: Thank you. We have revised the sentence to avoid this subjective term. The updated sentence now reads:

“Based on 5 cohorts, including over 2 million participants, the present study indicated that PPI use was associated with half of the top 30 diseases of global disease burden, with most of them exhibiting a dose-response relationship.”

10. Line 139: this is an observational setting that can at best estimate associations. The word effects suggest a causal relationship which is beyond the ambition on the current study. I suggest to replace the word effects throughout the manuscript with associations or relationships.

Response: Thank you for this suggestion. We have replaced the term “effects” with “associations” or “relationships” throughout the manuscript to accurately reflect the observational nature of the study and avoid implying a causal relationship.

11. Line 150: not really clear what a more comprehensive confounder control indicate. Please use less vague expressions.

Response: Thank you very much. We have revised the sentence for better clarity, as follows:

“Our study employed a more comprehensive approach to control for potential confounders. Specifically, we addressed confounding through 1) extensive adjustment for a wide range of covariates, 2) and the use of an active comparator in the CDARS database.”

12. Line 151: the sentence in item 3 does not read meaningful.

Response: Thank you. We have revised the sentence to enhance its readability:

“Study heterogeneity, reflecting variations in population characteristics, could also contribute to the disparities in findings.”

13. Line 154: avoid using the word effects.

Response: Thank you. We have replaced the word “effects” with “relationships” in the revised manuscript.

14. Line 155: replace diabetic risk with risk of developing type-2 diabetes.

Response: Thank you. We have replaced “diabetic risk” with “risk of developing type-2 diabetes” in the revised manuscript.

15. Line 156: a reference to the meta-analysis is missing.

Response: Thank you. There is no specific reference for this meta-analysis. Instead, we incorporated the results from four recent cohort or case-control studies (references 33-36), which were not included in the aforementioned meta-analysis, to update and strengthen our findings (Figure 1 below). The meta-analysis yielded a summary hazard ratio (HR) of 1.17 (95% CI: 1.01-

1.34), indicating a 17% increase in the risk of developing type-2 diabetes among PPI users compared to non-users, with high heterogeneity ($I^2=97\%$).

We have added the corresponding data to the statement in the revised discussion section, as follows:

“...A preliminary meta-analysis of all published results ($HR=1.17$, $95\%CI:1.01-1.34$, heterogeneity: $I^2=97\%$) suggested similar results as ours.”

Figure 1. Forest plot of PPI use and risk of developing type-2 diabetes based on aforementioned meta-analysis and four new studies using random-effects model.

16. Lines 157 to167: the text is potentially relevant but it is very dense but lacks specificity: it addresses multiple potential morbid conditions and candidate mechanisms at once.

Response: Thank you very much. We acknowledge that the text is dense and covers multiple potential morbid conditions and candidate mechanisms simultaneously. Because the underlying mechanisms is largely unclear and we have multiple potential morbid conditions, to describe mechanisms for individual outcomes will make it very complex and lead to redundancy. To enhance clarity and specificity, we have revised the section to offer a more focused and detailed account of the potential mechanisms underlying the observed associations between PPI use and the 15 specific diseases. Please find the updated explanation below:

“The intricate mechanisms linking PPI use to a spectrum of morbid conditions are multifaceted. By blocking acid production, PPI impairs one of the body’s natural defense mechanisms against ingested microorganisms, triggering profound changes in the gut microbiome.³⁹ This dysbiosis is evident in diarrheal diseases and involves the overgrowth of stomach bacteria, potentially increasing the risk of pneumonia through micro-aspiration.⁴⁰ The disturbance in the balance of microbial species in the gut and lungs may contribute to asthma through hyperactivation of T helper cell–dominated immune responses and the overproduction of inflammatory cytokines, leading to airway inflammation.⁴¹ Moreover, disruptions in the microbiome facilitate bacteria producing nitrosamines, and bile salt toxicity due to elevated stomach pH are potential

mechanisms for an increased risk of esophageal cancer.⁴² Enterococcus growth in the intestines translocating into the liver and inducing inflammation may contribute to chronic liver diseases.⁴³ Beyond the gastrointestinal realm, the derived hypomagnesemia and reduced insulin-like growth factor-1 (IGF-1) levels might facilitate diabetes mellitus development.⁴⁴ PPIs' impact on enteric infection, along with hypomagnesemia and uremic toxin accumulation, may contribute to CKD.⁴⁵ In neurological implications, small intestinal bacterial overgrowth (SIBO) and subsequent inflammatory responses are linked to Parkinson's disease.⁴⁶ PPI interference with the microbiome, hypergastrinemia, and potential impacts on central nervous system immune activity are suggested mechanisms for depressive disorders.⁴⁷ Beyond microbial effects, diminished gastric acidity in PPI users causes calcium or vitamin B12 malabsorption, decreasing bone mineral density and elevating the risk of osteoporosis and falls.⁴⁸ Additionally, PPIs may affect cardiovascular risk by modulating plasma asymmetric dimethylarginine, reducing nitric oxide levels, and impairing endothelium-dependent vasodilation.⁴⁹ It's essential to note that these findings are primarily derived from ex vivo studies, necessitating further investigation to elucidate the intricate associations observed.”

17. Line 170: what is indication for using PPIs?

Response: Thank you. The term “indication for using PPIs” refers to the medical reasons or conditions for which PPIs are prescribed or recommended. These indications include specific symptoms or diagnoses that lead to the prescription of PPIs, such as gastroesophageal reflux disease (GERD), gastric or duodenal ulcer, and gastrointestinal bleeding. In our study, we considered and adjusted for these indications when analyzing the associations between PPI use and various health outcomes to control for their potential confounding effects on the outcomes.

18. Line 171: add ... confounders .. in statistical models.

Response: Thank you. We have revised the sentence for clarity and grammatical correctness:

“To minimize confounding effects, our study first comprehensively adjusted for potential confounders in statistical models...”

19. Line 173: replace increased risk with positive associations.

Response: Thank you. We have replaced the term “increased risk” with “positive associations” in the revised manuscript.

20. Line 181: The sentence reading “our risk stratification model provides ...” is unclear. I wonder whether the Authors have a correct interpretation of what risk stratification represents. It is not a statistical model. Please consider revising the current version of the text.

Response: Thank you. We have revised the sentence to clarify the concept:

“Our risk stratification approach provides a feasible practical solution. Risk stratification, in this context, facilitates identifying individuals at higher or lower baseline risk for PPI-associated adverse events. This involves a comprehensive assessment of individual characteristics and health status through a prediction model. The importance of risk stratification is not only to identify those who are at high risk and take preventive measures individually to minimize the additional

harm caused by long-term PPI use, but also to screen patients who could safely use PPIs. This, in turn, reduces fears and increases treatment adherence among patients.”

21.in the paragraph: To the best of our knowledge, this is currently the most comprehensive assessment on the safety profile of PPIs. The outcome-wide approach allowed us to compare the effects of multiple outcomes, which reduced selective reporting bias. In addition, disease incidence information was ascertained by national record linkage or biennially updated information, which might reduce misclassification, recall bias, and attrition bias. Finally, the dose-response associations, robust sensitivity analyses as well as the negative control outcome (road injuries), increased our confidence in the accuracy of our findings.

The first sentence in Line 186: what is safety profile of PPIs?

Response: Thank you. The term “safety profile of PPIs” in the present study refers to the overall evaluation of the potential risks of 30 leading causes of global disease burden associated with the use of proton pump inhibitors (PPIs). We have revised the sentence to provide a more explicit explanation:

“To the best of our knowledge, this is currently the most comprehensive assessment on the long-term safety of PPIs, encompassing the associations with 30 leading causes of global disease burden...”

22. Line 191: Please consider revising the text, science is not about being confident about results.

Response: Thank you. We agree with you and we revised this sentence as follows:

“Finally, the dose-response associations, robust sensitivity analyses as well as the negative control outcome (road injuries), added additional strengths to our findings.”

23. Line 195: It is very informative that exposure consistency was evaluated across cohorts. What about exposure accuracy? In other words, was PPI exposure subject to misclassification. And if so, can one anticipate the level and the direction of bias?

Response: Thank you for this comment regarding exposure accuracy. We acknowledge that exposure misclassification is a potential limitation in our study, particularly in the UK Biobank cohort, where PPI use was self-reported at baseline without ongoing follow-up for changes in medication status.

In the NHS, NHS II, and HPFS cohorts, where information on PPI exposure was updated biennially, we used time-varying Cox regression to assess exposure, linking each assessment to subsequent events. This approach allowed for a more precise estimation of changes in exposure over the study period. In the CDARS database, detailed PPI prescription information, including drug names and duration, was recorded in the electronic data registration system, enhancing the accuracy of exposure assessment. For the UK Biobank, participants were initially assigned to the exposed or unexposed group based on self-reported PPI prescription status at enrollment. The lack of follow-up data on PPI usage introduces the possibility of misclassification, potentially underestimating the true effects, as the control group may include individuals who initiated PPI use during the follow-up period.

We reported this in the limitation of the revised manuscript as follows:

“Third, the definition of PPI use and endpoints is not always consistent among the included cohorts. Furthermore, PPI use was only evaluated once at baseline in the UK Biobank, introducing a chance of misclassification during follow-up. Misclassification could underestimate the true effects, as the control group may include individuals who initiated PPI use during the follow-up period. To minimize the potential influence, we combined the effects with random-effect model as other studies.^{7,8}”

24. Line 196: the use of random effects is not a limitation.

Response: Thank you. We apologize for the confusion caused by our previous wording. We did not mean use of random effects model is a limitation. We revised the limitation statement to avoid any misunderstanding:

“To minimize the potential influence, we combined the effects with random-effect model as other studies.^{7,8}”

25. Line 202 to 212: this is a useful paragraph. However, it is not obvious to understand how the results of the current study informed on the conclusions in this paragraph.

Response: Thank you very much. The comprehensive analysis of PPI use across multiple cohorts in our study provides essential information for understanding the implications for clinical practice and future research. By identifying associations between PPI use and a substantial global disease burden, our findings emphasize the importance of considering potential risks associated with long-term PPI use, even in the absence of established causal effects for these outcomes.

In practical terms, our findings highlight the need for personalized prevention strategies, particularly through the regular evaluation of baseline risk using readily available predictive factors for long-term PPI users. This approach is crucial, especially when focusing on high-risk patients, as the net risk of PPI-related adverse effects, although low in those with a low baseline risk, remains noteworthy due to the potential for changes in the baseline risk profile over time. For individuals at higher risk, potential effective strategies, such as dose reduction, discontinuation, transitioning to “on-demand” use, considering less profound acid suppressants like H2RAs, and regular monitoring for early indications of adverse events (e.g., blood glucose levels for the risk of diabetes), may help mitigate the additional absolute risk associated with PPI use.

However, it's crucial to note that our study does not exhaustively explore all aspects of PPI safety, and further research is warranted to address key areas. Specifically, ongoing research should aim to confirm causal effects through randomized controlled trial (RCT)-based meta-analysis, enhance and validate prediction models for various PPI-related adverse effects, determine appropriate cut-off values for defining high-risk populations, and evaluate the effectiveness of risk stratification strategies. These research avenues have the potential to refine clinical practices and optimize PPI use, ensuring a balanced approach between therapeutic benefits and potential risks in diverse patient populations.

We have also made appropriate revisions of this paragraph. Please see below:

“Implication for clinical practice and research

Given the links with a substantial global disease burden and the high rate of inappropriate overuse of PPIs (up to 70%),⁵¹ the potential impact of long-term PPI use should not be ignored, even if the causal effects for these outcomes have not been established. In practice, the net risk of PPI-related adverse effects is low in those with low baseline risk, but it is not negligible and the risk profile may change over time. Personalized prevention is feasible by regular evaluating the baseline risk with readily available predictive factors for long-term PPI users while focusing on the high-risk patients. For high-risk individuals, potential effective strategies, such as dose reduction, discontinuation, transitioning to “on-demand” use, considering less profound acid suppressants like H2RAs, and regular monitoring for early indications of adverse events (e.g., blood glucose levels for the risk of diabetes^{9,10}), may help mitigate the additional absolute risk associated with PPI use.

Further research is still required to 1) confirm the causal effects of PPIs on disease risk through RCT-based meta-analysis; 2) improve and validate the performance of prediction models for multiple PPI-related adverse effects; 3) investigate the appropriate cut-off value for defining high-risk population; 4) evaluate the effectiveness of the risk stratification strategy. These research avenues have the potential to refine clinical practices and optimize PPI use, ensuring a balanced approach between therapeutic benefits and potential risks in diverse patient populations.”

26. Line 216: the sentence on risk stratification reads unclear.

Response: Thank you. We have revised the sentence to enhance clarity:

“The risk stratification approach by individualized using of PPIs after evaluating the PPI-related risk, may be an effective strategy to reduce potential risks as well as fears among patients.”

27. Line 319: why were these participants excluded? It would be informative to model them.

Response: Thank you very much. The exclusion of participants who had used any PPIs two years before cohort entry was applied for several reasons.

Firstly, the repetitive use of PPI medications is a common practice in the population, and CDARS, being an electronic database containing health records from nearly all public hospitals and clinics in Hong Kong, captures information about individuals' illnesses and their repeated medication use. The exclusion criterion was applied to eliminate the residual effects of previously used PPIs, particularly for those engaging in intermittent PPI use. This approach helps ensure a more accurate assessment of the impact of new PPI use on the outcomes of interest.

Additionally, implementing a two-year washout period serves as a practical strategy to distinguish new PPI users from those with a history of regular PPI use. It minimizes the potential carryover effects or confounding influences from past long-term PPI exposure, providing a clearer delineation of the association between recent PPI use and the investigated health outcomes. This

helps enhance the internal validity of the study by focusing on a cohort of participants with more relevant and recent PPI exposure.

Therefore, the exclusion of participants with a history of PPI use two years before cohort entry was a methodological choice to better isolate and assess the impact of new regular PPI use on the outcomes under investigation.

28. Line 360: Please specify what was the primary time variable in Cox models.

Response: Thank you. The primary time variable in the Cox models for the risk stratification model was the time from UK Biobank entry to the occurrence of any of the 15 PPI-related diseases, death, or loss to follow-up, whichever comes first.

We have added the corresponding information in the revised manuscript as follows:

“...The primary time variable considered in this analysis was the duration from UK Biobank entry until the first instance of any of the 15 PPI-related diseases, death, or loss to follow-up, whichever came first...”

29. Line 361-362. The strategy used does not comply with a formal definition of confounders. Confounders also need to show some association with the main exposure(s).

Response: Thank you.

The aim of this part is not to control confounding, but to identify predictors. Predictors do not necessarily have to be non-confounders. Therefore, it is not necessary for these potential predictors to show an association with the corresponding exposure.

We revised the text to more accurately reflect this:

“We utilized Cox regression models to identify potential predictors associated with the occurrence of any of the 15 PPI-related diseases, aiming to construct a comprehensive predictive model and estimate coefficients for each identified risk factor...”

30. in the data analysis, the author said: To address potential reverse causation (i.e., symptoms of undiagnosed diseases resulting in PPI prescription), we performed a time-lagged analysis of the exposure for 2 years, which could strengthen the temporality and allow a time window for disease risk development.

Line 375: the lagged analyses does not read clear enough.

Response: Thank you. We have revised the description of the time-lagged analysis for clarity. The updated version is as follows:

“To address potential reverse causation (i.e., symptoms of undiagnosed diseases resulting in PPI prescription), our analyses were restricted to patients with at least 2 years of follow-up after cohort entry, introducing a 2-year exposure-lag period. This approach aims to strengthen the

temporality of our analysis by allowing for a sufficient latency period for disease risk development, while also minimizing the impact of detection bias.”

31. In tables and figures, please make more informative use of footnotes to clarify important features of the quantities reported.

Response: Thank you. We have revised the tables and figures to include more informative footnotes, providing clarification on important features of the reported quantities. This may improve the reader's understanding of the presented data.

32. Table 1: postmenopausal (%), specify ... among women?

Response: Thank you. “postmenopausal (%)” in Table 1 refers to the percentage of postmenopausal women for each cohort. This information has been specified in the updated table.

33. T1: add alcohol to “Never drinkers”

Response: Thank you. We have replaced “Never drinkers” with “No-alcohol drinkers” in the revised table 1.

34. T1: Comorbidities: Are these prevalent conditions? Please clarify.

Response: Thank you. The term “Comorbidities” in Table 1 refers to prevalent conditions. We have clarified this by explicitly stating “Prevalent comorbidities” in the revised table to enhance clarity for readers.

35. T1: Please add information about PPI frequency and duration.

Response: Thank you. Unfortunately, we couldn't include information about PPI frequency and duration in Table 1 as the UK Biobank and the three U.S. cohorts (NHS/NHS II/HPFS) lack data on detailed PPI usage. The data for PPI duration in this study are derived from follow-up assessments in the CDARS database, specifically focusing on new PPI users defined as those who used PPIs for >30 days within the 2-year cohort entry period. Therefore, it is not feasible to adequately reflect these details in the baseline demographic characteristics presented in Table 1. Instead, we have included the proportion of PPI users in each cohort in the revised Table 1.

36. Figure 1, the flowchart: it is not entirely readable.

Response: Thank you. We have revised Figure 1 to enhance its readability. We have included information on the number of incident chronic conditions in each cohort as you have suggested, providing a more comprehensive overview. Please find the updated Figure 1 below:

Figure 1. Flowchart of participant inclusion.

37. F1: replace baseline for basic (population).

Response: Thank you. We have replaced “basic (population)” with “baseline” in Figure 1 to accurately reflect the information.

38. F1, the blue box: the reported N should indicate the number of incident chronic conditions, rather than the participants for specific events.

Response: Thank you. In the revised figure 1, we have included information on the number of incident chronic conditions in each cohort as suggested.

39. F1: frequencies reported here are not consistent with T1.

Response: Thank you. We have carefully reviewed the data, and upon further examination, we found that the baseline participants in Figure 1 are consistent with the total numbers reported in Table 1. The original discrepancy in frequencies may be due to Figure 1 illustrating the inclusion and exclusion of participants for each specific disease cohort, whereas Table 1 provides an overview of the baseline demographic characteristics for the entire study population. We appreciate your diligence in reviewing the details, and we can confirm the accuracy of the presented information.

REVIEWERS' COMMENTS

Reviewer #1 (Remarks to the Author):

All my prior comments and questions have been appropriately addressed and were incorporated in the manuscript if appropriate. I do not have any further comments.

Reviewer #2 (Remarks to the Author):

I think the manuscript is ready now to be published

Reviewer #4 (Remarks to the Author):

The Authors accurately reviewed their manuscript according to the feedback received from the various Reviewers. No comments to add